# The RNA m⁶A landscape during human oocyte-to-embryo transition

Yanjiao Li[1,2,7], Yunhao Wang [ID][3,7], Aylin Cengiz[1,2,7], Kang-Xuan Jin[1,2], Blanca Corral Castroviejo[4], Xiaolin Lin [ID][1,2,5], Marie Indahl[2,4,6], Rujuan Zuo[1], Trine Skuland [ID][2,4,6], Madeleine Fosslie [ID][1,2], Maria Biba[2,4,6], Xuechen Wu[1,2], Peter Fedorcsak [ID][2,4,6], Magnar Bjørås [ID][1,2,5], Adam Filipczyk[1], John Arne Dahl [ID][1,2✉], Gareth D Greggains [ID][1,2,4✉], Kin Fai Au [ID][3✉] & Arne Klungland [ID][1,2✉]

## Abstract

RNA *N*⁶-methyladenosine (m⁶A, m6A) modification is a critical regulator for a range of physiological processes. However, the dynamic m⁶A profiles within human preimplantation embryos remain uncharacterized. Here, we present the first RNA m⁶A landscape of single human oocytes and early embryos. Comparative analyses with mouse data reveal an intriguing divergence during the window of zygotic genome activation. m⁶A-modified genes are involved in regulation of gene transcription, while unmodified genes are mainly associated with basic metabolic processes. Maternal decay mRNAs exhibit a propensity for m⁶A modifications, and these genes are targeted by miRNAs. m⁶A modified genes that are constantly expressed across all stages demonstrate higher translation efficiency. Moreover, we observe frequent m⁶A enrichment on stage-specifically expressed retrotransposons, particularly within young subfamilies. m⁶A inhibitor leads to m⁶A erasure on massive retrotransposons. In summary, this study provides a resource to broaden our understanding about the regulatory roles of m⁶A during early human embryo development.

**Keywords** m6A; Human Early Embryos; Zygotic Genome Activation; Retrotransposons
**Subject Categories** Development; RNA Biology

## Introduction

Human infertility is a growing global concern, with one in six individuals worldwide affected by this condition (Cox et al, 2022). Understanding the processes of human oocyte maturation and early embryonic development is crucial to deciphering its etiology.

RNA modifications have emerged as pivotal regulators of gene expression programs (Shi et al, 2019; Yang et al, 2018). In mammals, the most prevalent internal mRNA modification is *N*⁶-methyladenosine (m⁶A, m6A) (Frye et al, 2018; Roundtree et al, 2017). m⁶A modification plays crucial roles in various biological processes related to reproduction and fertility (Klungland et al, 2017; Zheng et al, 2013). The key role of m⁶A dynamics in meiosis seems to be well conserved and was identified early in pioneering studies in yeast (Schwartz et al, 2013; Shah and Clancy, 1992). We and others have characterized the dynamic m⁶A modification in mouse oocytes and early embryos, uncovering its important functions in early embryonic development (Wang et al, 2023; Wu et al, 2022; Yao et al, 2023; Zhu et al, 2023). However, no study has yet investigated this crucial RNA modification in human early embryos. This is primarily due to two major challenges: first, human embryos are precious and difficult to obtain as research materials, as their use is strictly regulated by ethical constraints; second, various methods have been developed to identify transcriptome-wide m⁶A profiles, significantly advancing our understanding of m⁶A (Chen et al, 2015a; Dominissini et al, 2012; Garcia-Campos et al, 2019; Hu et al, 2022; Linder et al, 2015; Liu et al, 2023; Meyer, 2019a; Meyer et al, 2012; Pratanwanich et al, 2021; Shu et al, 2020; Wang et al, 2020; Xiao et al, 2023; Zhang et al, 2019), however, given the scarcity of human early embryos, none of these methods can be directly applied to this context, as they would require thousands of embryos. Recently, two unique droplet-based methods, scDART-seq and m6A-CT, have been developed to enable m⁶A profiling of single cells, however, both require thousands of cells as starting materials (Hamashima et al, 2023; Tegowski et al, 2022). Another method, scm⁶A-seq, which barcodes individual embryos, enabled the first m⁶A profiling of early mouse embryos by using multiple embryos as the starting material for each developmental stage (Yao et al, 2023). Recently, we developed picoMeRIP-seq to study m⁶A using single oocyte or embryo as starting material (Li et al, 2024). picoMeRIP-seq opens the door for m⁶A mapping in vivo in single cells and scarce cell types. Earlier studies have shown that reprogramming of DNA methylation is

[1]Department of Microbiology, Oslo University Hospital, Rikshospitalet, Oslo, Norway. [2]Centre for Embryology and Healthy Development, University of Oslo, Oslo, Norway. [3]Gilbert S. Omenn Department of Computational Medicine and Bioinformatics, University of Michigan, Ann Arbor, MI, USA. [4]Department of Reproductive Medicine, Oslo University Hospital, Rikshospitalet, Oslo, Norway. [5]Department of Clinical and Molecular Medicine (IKOM), Norwegian University of Science and Technology (NTNU), 7491 Trondheim, Norway. [6]Division of Gynaecology and Obstetrics, Institute of Clinical Medicine, Faculty of Medicine, University of Oslo, Oslo, Norway. [7]These authors contributed equally: Yanjiao Li, Yunhao Wang, Aylin Cengiz. ✉E-mail: j.a.dahl@medisin.uio.no; g.d.greggains@ous-research.no; kinfai@umich.edu; arne.klungland@medisin.uio.no

highly similar between mouse and human early embryos (Guo et al, 2014; Smith et al, 2014). Understanding whether the m⁶A landscapes in early embryonic RNAs are conserved between humans and mice is essential for comprehensive insights on RNA m⁶A methylation. Here, we provide a valuable resource on the RNA m⁶A landscape during the human oocyte-to-embryo transition. To the best of our knowledge, this is the first study to explore transcriptome-wide RNA modifications in human oocytes and early embryos, deepening our understanding of m⁶A's regulatory roles in early human embryo development.

## Results

We recently reported a highly sensitive picogram-scale m⁶A RNA immunoprecipitation and sequencing (picoMeRIP-seq) method that can start out with a single mouse oocyte, or a single mouse or zebrafish early embryo (Li et al, 2024). Here, we applied picoMeRIP-seq to human oocytes and early embryos to profile the m⁶A landscape and to identify potential regulators during human oocyte-to-embryo transition (OET) (Fig. EV1A; "Methods").

First, we demonstrated that picoMeRIP-seq could acquire high-quality data from as few as 10 human embryonic stem cells (hESCs) (Fig. EV1B–F). As shown in genome browser, a high level of reproducibility for selected gene loci was observed (Fig. EV1B). Transcriptome-wide profiles of picoMeRIP-seq data from 1000 cells to 10 cells showed strong correlations between replicates and across different input cell numbers (Pearson correlation coefficients >0.8; Fig. EV1C). A high percentage of genes with m⁶A identified in samples of 10 cells showed an overlap with those identified in samples of 1000 cells (>85%), and with those reported in previously published m⁶A-seq data (>75%) (Batista et al, 2014) (Fig. EV1D). In agreement with previous studies (Batista et al, 2014; Dominissini et al, 2012; Meyer et al, 2012), m⁶A peaks were predominantly enriched around stop codons (Fig. EV1E), and the most significant consensus motif within m⁶A peaks was GGACU (Fig. EV1F). These results highlight the efficacy of picoMeRIP-seq in accurately profiling the m⁶A methylome from low-input human samples, demonstrating its strong sensitivity and reproducibility.

Next, we performed picoMeRIP-seq on single human oocytes at the germinal vesicle (GV) and metaphase II (MII) stages, and single human embryos at the zygote (1C), two-cell (2C), eight-cell (8C) and blastocyst (BLT) stages, with ≥2 biological replicates for each stage (Table EV1). Principal component analysis (Fig. EV1G) and hierarchical clustering analysis (Fig. 1A) of the transcriptome-wide m⁶A profiles showed the precise clustering of single oocytes and embryos by cell identity. For the 2C and 8C stages, we had only two biological replicates available. To maintain statistical rigor and ensure comparability between stages, we selected two biological replicates from each stage for downstream analyses. The number of m⁶A peaks identified ranged from 5401 to 9601 across the various stages (Fig. EV1H). We identified 5126 (GV), 4497 (MII), 3666 (1C), 5328 (2C), 4820 (8C) and 5519 (BLT) m⁶A-modified (m⁶A+) genes (Fig. 1B; Dataset EV1). Among highly expressed genes (TPM ≥ 10), the percentage of m⁶A+ genes ranged from 32 to 44%, aligning with the overall trend observed in the number of m⁶A+ genes across the six stages (Fig. EV1I). Overall, the numbers and the percentage of m⁶A+ genes decreased until fertilization and then

increased after the zygote stage, the minimal number of m⁶A-modified genes at the 1C stage could result from significant maternal RNA degradation before substantial zygotic mRNA synthesis begins, which suggests that m⁶A is dynamically regulated during the human preimplantation embryos. This could be credited to the dynamic expression of the m⁶A methyltransferases METTL3 and METTL14, the m⁶A demethylases FTO and ALKBH5, and m⁶A readers at both RNA and protein levels (Yang et al, 2018; Zheng et al, 2023; Zhu et al, 2025) (Figs. EV1J and EV2A). Among these m⁶A+ genes, the majority were protein-coding genes (82–92%), but there were also long non-coding RNAs (lncRNAs, 5–12%), pseudogenes (2–6%) and a small fraction of other types of RNA species (<1%) (Fig. EV2B,C). The non-coding m⁶A + RNAs identified at each stage in this study may play potential regulatory roles during human oocyte and early embryonic development, as demonstrated by previous study that established the m⁶A-associated functions of lncRNAs in human fetal tissues (Xiao et al, 2019). For instance, m⁶A has been demonstrated to be required for the function of *XIST* (Patil et al, 2016), which is a lncRNA that mediates gene silencing on the X chromosome during mammalian development and was marked by m⁶A at the BLT stage (Fig. 1C). The m⁶A profiles were clearly detectable as peaks, as illustrated by genome browser snapshots of representative genes, such as oocyte-secreted factor *GDF9* (Paulini and Melo, 2011), the 8C embryo marker gene *LEUTX* (Lewin et al, 2023), blastocyst-specifically activated gene *ARID3A* (Rhee et al, 2014), and an imprinted gene *ZDBF2* that was constantly expressed across all stages (Duffie et al, 2014) (Fig. 1C). The m⁶A peaks typically exhibited significant enrichment near the stop codons (Figs. 1D,E and EV2D), and displayed the consensus motif GGACU (Figs. 1F and EV2E) across all stages.

We further explored the extent to which differences in the m⁶A methylome during OET are conserved between human and mouse by comparing their transcriptomes and m⁶A methylomes from equivalent stages (Wang et al, 2023) (Fig. 2A). To mitigate the influence of RNA abundance on m⁶A characterization in the two species, we specifically targeted genes with high expression levels (TPM ≥ 50) at each stage in either species, noting that m⁶A+ genes had median TPM around 50 (Fig. EV3A). For the non-homologous genes between the two species, an average of 39% were m⁶A+ across all stages in human, compared to 49% in mouse, with a notably low fraction of m⁶A modification at the BLT stage for both human (20%) and mouse (35%) (Fig. EV3B).

Next, we focused on homologous genes and categorized them into three groups based on their expression at each stage: human-mouse co-expressed (expressed in both human and mouse), human-specifically expressed, and mouse-specifically expressed genes (Fig. 2B; Dataset EV2; "Methods"). Among human-mouse co-expressed genes, approximately 35% were m⁶A+ in both species across various stages, with <11% uniquely modified in human while >17% uniquely modified in mouse (Fig. 2B, middle). Notably, 36% of co-expressed genes at the 8C/2C stage were exclusively modified at the mouse 2C stage versus only 7% at the human 8C stage. Human major zygotic genome activation (ZGA) occurs at the eight-cell stage, whereas in mice it takes place at the two-cell stage. This observed divergence could be attributed to differences in developmental timing or the distinct profiles of major zygotic gene expression, comparisons of human and mouse transcriptomes during the major ZGA show that only half of the transcriptomes

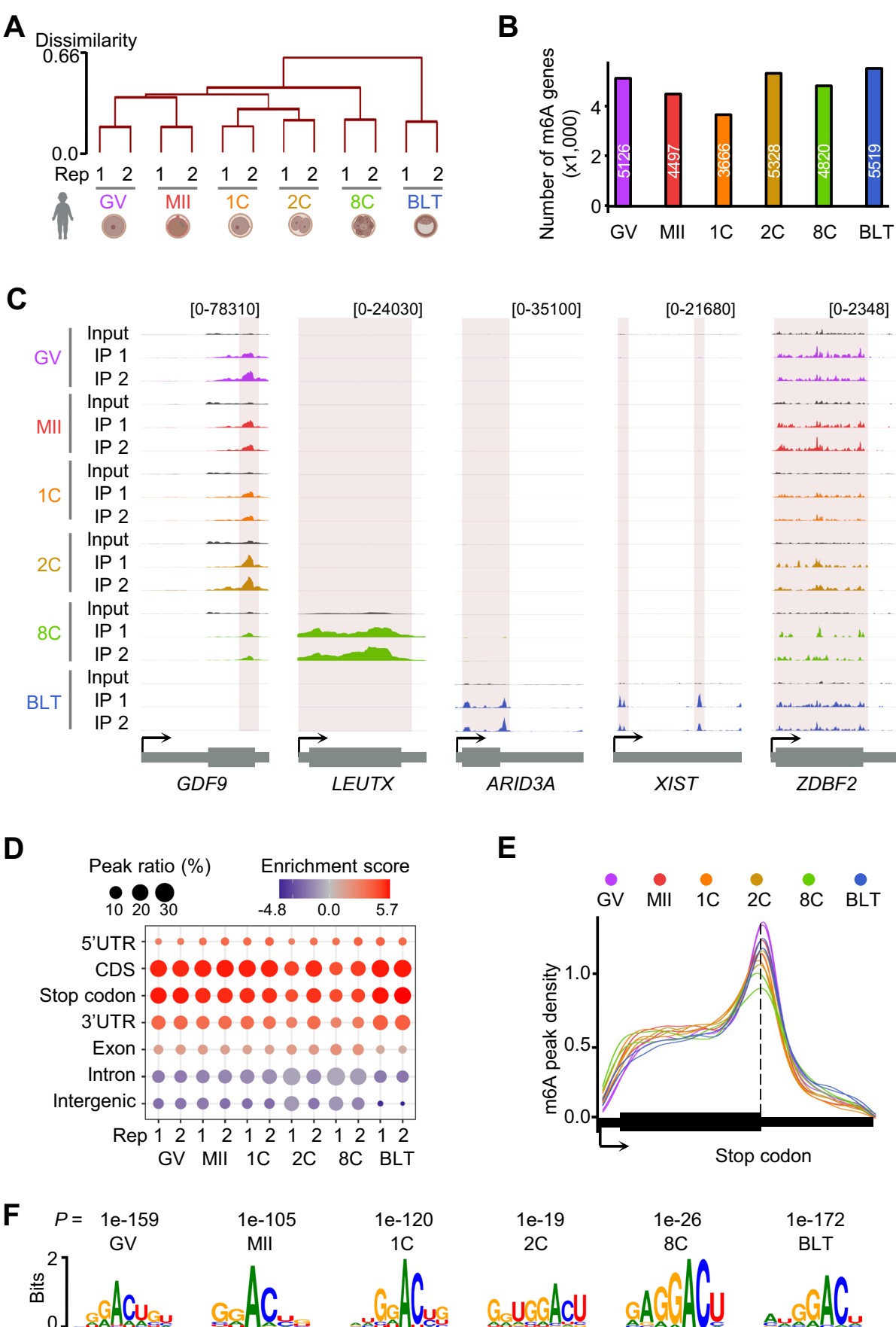

**Figure 1. Dynamic RNA m⁶A landscape in human oocytes and preimplantation embryos.**

(A) Hierarchical clustering analysis of transcriptome-wide m⁶A profiles. (B) Number of m⁶A+ genes. (C) Representative genome browser view of picoMeRIP-seq signals. (D) Bubble plot representing the relative enrichment ratios in different genomic features. For each feature, the enrichment score was calculated as the $\log_2$ ratio of the observed peak numbers to the expected peak numbers. (E) Metagene profiles of m⁶A peak distribution. (F) Consensus motifs on m⁶A peaks identified in biological replicate 1.

overlap, suggesting that the homology between human and mouse zygotic genes is relatively low (Sha et al, 2020). This disparity is further reflected in the genes that are specifically expressed in either human or mouse. Regarding human-specifically expressed genes, over 50% were m⁶A+ at the GV, MII and BLT stages, whereas this decreased to 35% at the 8C stage (Fig. 2B, top). In contrast, in mouse-specifically expressed genes, >54% were m⁶A+ at every stage, with a peak of 65% at the 2C stage (Fig. 2B, bottom). These data reveal distinct RNA m⁶A methylome states during the human major ZGA, likely reflecting evolutionary divergence.

Recent advances in understanding m⁶A deposition mechanisms indicate that m⁶A acts like a default hardware feature (He et al, 2023; Uzonyi et al, 2023; Yang et al, 2022). These studies found that longer genes tend to exhibit higher m⁶A levels, with the exon junction complex (EJC) acting as an m⁶A suppressor to shape the m⁶A epitranscriptome, protecting RNA near exon junctions within coding sequences from methylation. Also, longer genes typically have more exons, which increases the availability of RRACH (R = G or A; H = A, C or U) motifs in exonic regions. We aim to determine if this finding represents a universal mechanism or context-dependent differences in human early embryonic development. For both human- and mouse-specifically expressed genes, m⁶A+ genes exhibited significantly longer gene lengths than unmodified genes across all stages (Figs. 2C and EV3C). In human-mouse co-expressed genes, m⁶A+ genes in both species were substantially longer than others; and interestingly, human homologous m⁶A+ genes were significantly shorter than mouse m⁶A+ homologous genes specifically at the 2C and 8C stage, respectively (Fig. 2C), a distinction not observed at other stages (Fig. EV3C). Again, this could be attributed to the unequal cell stage comparison between the mouse 2C and human 8C stages. m⁶A+ genes contained markedly more RRACH motifs than unmodified genes across all stages; and for human genes co-expressed in mouse, the genes were uniquely expressed in m⁶A + mRNAs in human had lower counts of RRACH motifs than those uniquely expressed in m⁶A + mRNAs in mouse, specifically in human 8C embryo, likely due to the unequal cell stage comparison (Figs. 2D and EV3D). It is important to note that, after normalizing for gene length, there is no significant difference in the density of RRACH motifs (Fig. EV3E). Instead, this may be more closely related to gene length and splicing events, as supported by recent studies (He et al, 2023; Uzonyi et al, 2023; Yang et al, 2022). These analyses indicate that our data quality is consistent with the latest findings in the field, demonstrating that picoMeRIP is a highly accurate method for RNA m⁶A profiling.

Gene ontology (GO) analysis provided functional insights into different groups of genes: of human-specifically expressed genes at the GV, MII, 1C and 8C stages, m⁶A+ genes were largely linked to transcription regulation while unmodified genes were mainly enriched in basic metabolic processes; at the BLT stage, human-specifically expressed m⁶A+ genes contributed to cell differentiation (Fig. 2E). Additionally, for mouse-specifically expressed m⁶A+ genes, divergence in function by stage was observed: they were implicated in the regulation of cell proliferation and multicellular organism development from GV to the 1C stage, and in transcription regulation from GV to the 2C stage (Fig. EV4A). Across all stages, for human-mouse co-expressed genes, m⁶A+ genes were largely associated with transcription, embryonic development as well as development-related signal pathways, while unmodified genes were frequently involved in diverse metabolic pathways (Fig. EV4B,C). Taken together, these results underscore the conserved characteristics of m⁶A modification between the two species. The observed divergences between human 8C and mouse 2C embryos suggest the possibility of species-specific roles for m⁶A during the critical window of major ZGA.

We sought to further focus on m⁶A profiling of the two critical events during OET: maternal RNA degradation (MD) and ZGA (Fig. 2A). We classified MD genes into two subsets based on their distinct decay patterns: M-decay, degraded during oocyte maturation, exemplified by *VCAN* that is associated with human oocyte developmental competence (Shen et al, 2020); and Z-decay, degrading after fertilization, such as *BNC1* that is required for human oocyte meiosis (Zhang et al, 2018) (Fig. 3A,B; Dataset EV3; "Methods"). In total, we identified 1339 MD genes (M-decay, 396; Z-decay, 943) and 550 ZGA genes (Fig. 3A). The smaller number of m⁶A-marked genes detected in this study compared to mouse studies is likely influenced by the larger number of embryos (at least hundreds of embryos at each stage) used in mouse studies, which allows for greater enrichment of low-abundance RNAs (Wang et al, 2023; Wu et al, 2022; Zhu et al, 2023). Near half of ZGA genes were methylated at the 8C stage (Fig. 3C), including *TFAP2C*, a known marker of human ground-state naive pluripotency (Pastor et al, 2018) (Fig. 3B). Notably, some MD genes were consistently tagged with m⁶A throughout development. We hypothesize that although the original maternal mRNAs are degraded, the zygotic genome re-expressed new mRNAs from the same genes, ensuring continued support for embryonic development (Sha et al, 2020; Yan et al, 2013), consistent with the results in mouse (Wang et al, 2023; Wu et al, 2022). In contrast to M-decay genes (<19%), Z-decay genes (>30%) displayed significantly higher m⁶A+ proportions at the GV, MII and 1C stages (Fig. 3C), which can be partially explained by the relatively lower expression of M-decay genes (Fig. EV5A). For M-decay genes at the GV stage, Z-decay genes at the GV, MII and 1C stages, as well as ZGA genes at the 8C stage, those with m⁶A modification had significantly longer gene lengths (Figs. 3D and EV5B) and more RRACH motifs (Figs. 3E and EV5C) than unmodified genes. Unmodified MD genes were predominantly associated with metabolic pathways, apoptotic process and protein transport, whereas unmodified ZGA genes were linked to RNA processing and nucleosome assembly (Fig. 3F). Of note, m⁶A+ genes from both MD and ZGA functioned

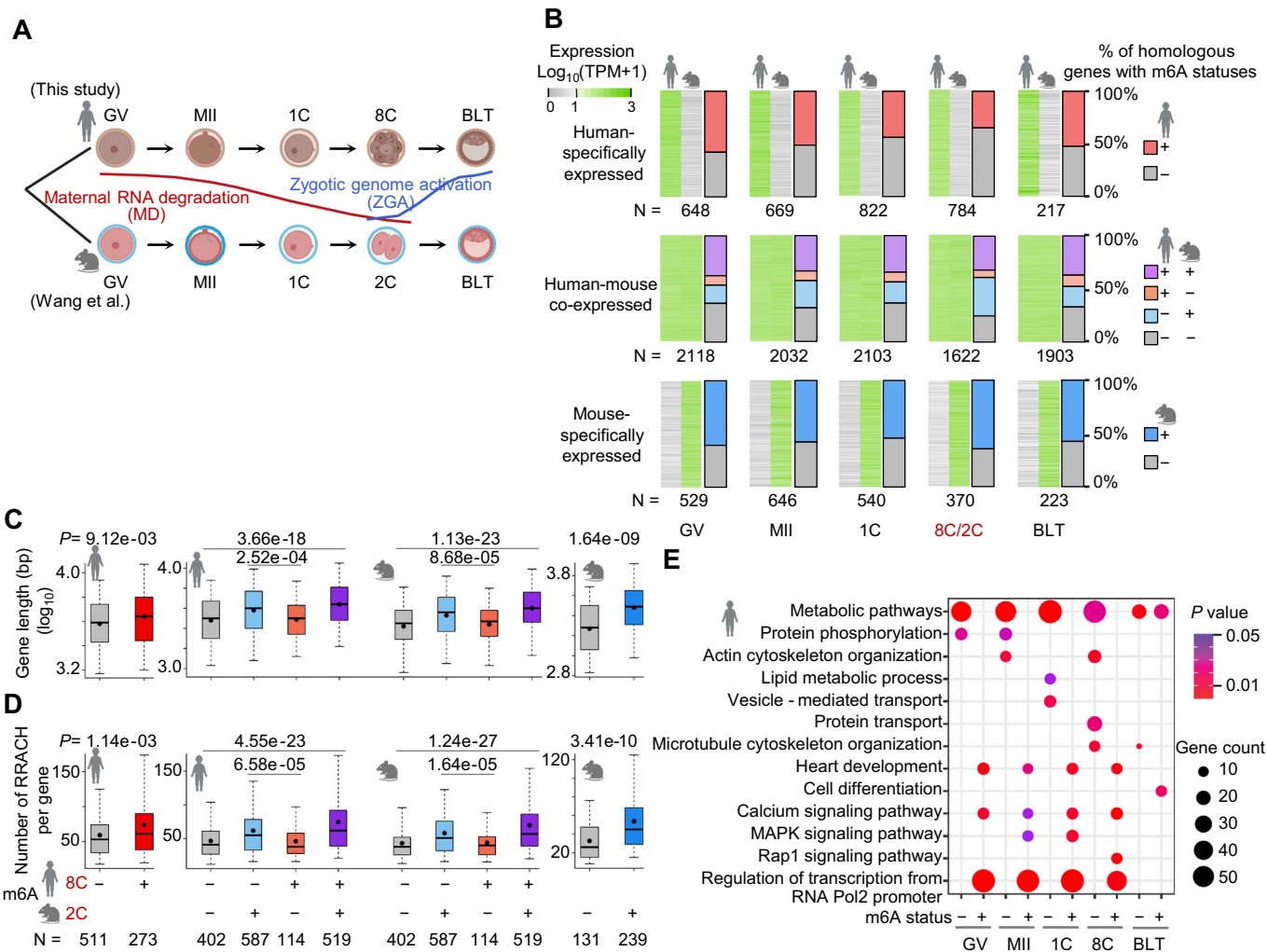

**Figure 2. Conservation and divergence of m⁶A methylation during human and mouse OET.**

(A) Schematic of human and mouse oocytes and embryos used for stage comparison. Human ZGA mainly occurs at the 8C stage, while mouse ZGA occurs at the 2C stage. (B) Heatmaps showing the expression levels (left) of three categories of homologous genes in humans and mice for each developmental stage; and the ratios of genes with different m⁶A statuses (right). (C, D) Comparison of gene lengths (C) and RRACH counts (D) between m⁶A modified and unmodified homologous genes at the human 8C and mouse 2C stages. The P values were calculated by the two-sided Wilcoxon rank-sum test. The boxplots are generated with five percentile-based whiskers, arranged from top to bottom as follows: the 95th percentile, the 75th percentile, the median, the 25th percentile, and the 5th percentile; and the mean value indicated by a dot. N number of genes used in the data analysis. (E) GO analysis for m⁶A modified and unmodified human-specifically expressed genes. Fisher's exact test was used to calculate the one-sided P values.

in regulation of cell proliferation and pluripotency, likely mediated by key factors essential for transcription (Fig. 3F). Given that m⁶A is believed to regulate cell fate in embryonic stem cells by modulating its abundance on transcription factor coding genes (Batista et al, 2014; Geula et al, 2015; Jin et al, 2021), we examined the expression and m⁶A modification dynamics of genes encoding transcription factors (Fig. 3G) and transcription cofactors (Fig. EV5D) throughout all six stages. Among the highly expressed genes (TPM ≥ 50), an average of 68% of transcription factors and 48% of transcription cofactors (Fig. EV5E) were m⁶A+ across all stages, with the highest methylation ratios at the BLT stage. The mature human blastocyst is comprised of the first three cell lineages of the embryo: trophectoderm, epiblast and primitive endoderm (Alberio, 2020; Blakeley et al, 2015; Stirparo et al, 2018). We further

examined the m⁶A status on the lineage-specific marker genes across developmental stages and observed m⁶A occupancy on some master transcription regulators at the 8C and/or BLT stages, such as *GATA2*, *GATA3* and *TEAD3* (markers of trophectoderm), *SOX2* and *NANOG* (markers of epiblast), and *PDGFRA* and *GATA6* (markers of primitive endoderm) (Fig. EV5F). The prevalence of m⁶A deposition on maternal RNAs and the onset of new m⁶A methylation during ZGA, particularly on those genes involved in transcription regulation, hint at the involvement of m⁶A in these two critical events during human OET.

MicroRNAs (miRNAs), along with m⁶A, are another key regulator of RNA stability and translation, and are essential for mammalian development (DeVeale et al, 2021). We and others have observed that m⁶A modifications are particularly enriched at

miRNA targeting sites in various human and mouse tissues (Liu et al, 2020) as well as in mouse pluripotent stem cells and early embryos (Chen et al, 2015b; Wang et al, 2023). Next, we examined

the co-occupancy between m⁶A modification and miRNAs on human MD and ZGA genes (Dataset EV4; "Methods"). miRNAs had a significant tendency to target m⁶A + Z-decay genes as

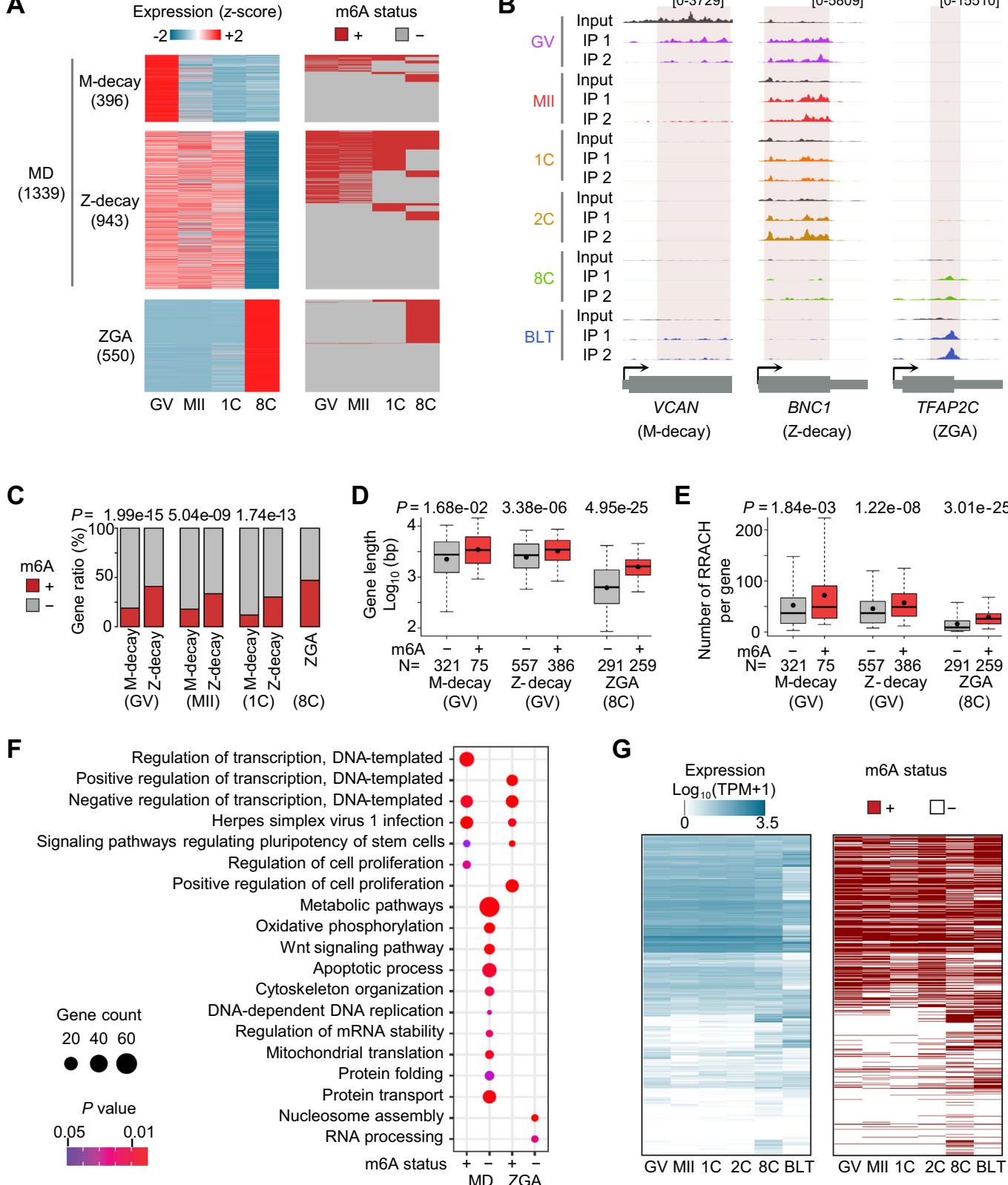

**Figure 3. Enrichment of m⁶A in MD and ZGA genes, as well as in transcription factor coding genes.**

(A) Expression (left) and m⁶A modification (right) dynamics of M-decay, Z-decay and ZGA genes. (B) Genome browser views showing the m⁶A profiles of representative M-decay, Z-decay and ZGA genes. (C) Percentage of m⁶A+ genes for each event at each specified stage. The *P* values are calculated by the one-sided Fisher's exact test. (D, E) Comparison of gene lengths (D) and RRACH counts (E) between m⁶A modified and unmodified genes for M-decay, Z-decay and ZGA genes. The *P* values were calculated by the two-sided Wilcoxon rank-sum test. The boxplots were generated with five percentile-based whiskers, arranged from top to bottom as follows: the 95th percentile, the 75th percentile, the median, the 25th percentile, and the 5th percentile; and the mean value indicated by a dot. *N* number of genes used in the data analysis. (F) GO analysis for m⁶A modified and unmodified genes for MD and ZGA genes. Fisher's exact test was used to calculate the one-sided *P* values. (G) Expression (left) and m⁶A status (right) dynamics of transcription factor coding genes.

compared with unmodified genes at the GV, MII and 1C stages (Figs. 4A and EV6A,B). However, unlike our observations in mouse data (Chen et al, 2015b; Wang et al, 2023), we did not discern a distinct preference of miRNA targeting between modified and unmodified M-decay genes at the GV stage and ZGA genes at the 8C stage (Figs. 4A and EV6A,B). Previous studies have demonstrated significant differences in gene expression and miRNA profiles between humans and mice during oocyte maturation and major ZGA. We reasoned that mice and human may have divergent mechanisms for miRNA regulation, which likely explains the distinct miRNA targeting phenomenon observed here (Paloviita et al, 2021; Sha et al, 2020; Yang et al, 2016; Zhao et al, 2020). This discrepancy may suggest species-specific differences in the coordination of m⁶A modification and miRNA targeting during OET between humans and mice. However, when examining genes that were constantly expressed across all stages, the preference of miRNA targeting on m⁶A+ genes was substantially higher than on unmodified genes (Figs. 4B and EV6C,D).

m⁶A is known to regulate RNA translation efficiency (Jain et al, 2023; Shan et al, 2023; Wang et al, 2015). By correlating previous translatome data in human oocytes and early embryos (Zou et al, 2022) with our m⁶A methylome data, we found that m⁶A + Z-decay genes had relatively higher translation efficiency than unmodified genes at the MII and 1C stages, while there was no significant association between m⁶A modification and mRNA translation efficiency for M-decay and ZGA genes (Fig. 4C; Dataset EV5; "Methods"). For constantly expressed genes, the median translation efficiencies of m⁶A+ genes were higher than unmodified genes across different stages (Fig. 4D). The locations of m⁶A deposition along mRNAs are closely related to its distinct roles in translation control (Meyer, 2019b; Shan et al, 2023). We categorized constantly expressed m⁶A+ genes based on their modification sites into 5'UTR, coding sequence (CDS), stop codon and 3'UTR. Despite previous reports of a negative correlation between CDS m⁶A and translation in other research models (Meyer, 2019b; Shan et al, 2023), our observations showed that genes with CDS m⁶A residing had relatively higher translation efficiencies in comparison to those with m⁶A at other locations, especially in oocytes (Fig. 4E). The role of CDS m⁶A in either promoting or inhibiting protein production remains elusive (Meyer, 2019b). The variations in m⁶A associated translation efficiency between MD and ZGA events, as well as the divergent impacts of CDS m⁶A on translation between our observation and previous studies, suggest that the effect of m⁶A on translation is multifaceted and complex. It possibly involves different m⁶A readers to modulate translation positively or negatively under different conditions (Meyer, 2019b; Shan et al, 2023).

Recent findings showed that the m⁶A modified transcripts derived from retrotransposons could act both *in cis* and, in trans to regulate the

chromatin environment, hence, the m⁶A modification has been linked to cell fate determination through modulating retrotransposon-derived RNAs (Chelmicki et al, 2021; Chen et al, 2021; Liu et al, 2021; Sun et al, 2023; Xu et al, 2021). Here, we found that m⁶A was largely enriched on retrotransposon-derived RNAs during human OET (Fig. 5A; Dataset EV6; "Methods"). The most recently integrated human endogenous retrovirus H (HERVH) were consistently activated across all stages, and >51% of the expressed HERVH loci/copies were modified by m⁶A from the GV to 2C stages (Fig. 5A). HERVK showed marked m⁶A modification at the BLT stage. In addition to full-length endogenous retroviruses (ERVs), solo long terminal repeats (LTRs), such as LTR12C and THE1C, were also heavily occupied by m⁶A prior to ZGA (Fig. 5A). Certain evolutionarily young subfamilies of LINE-1, including L1HS, L1PA2, and L1PA3, demonstrated their highest m⁶A frequencies at the BLT stage. SVA elements also exhibited a high degree of m⁶A modification concurrent with ZGA. There was no frequent enrichment of m⁶A on Alu elements. Notably, HERVH displayed a broad distribution of m⁶A across the internal sequences, while solo THE1C exhibited a clear positional preference for m⁶A occupancy (Figs. 5B and EV6E). Further examination of other types of retrotransposons supports this divergent distribution of m⁶A along their entire sequences (Fig. EV6E). Due to their highly repetitive nature, one of the technical challenges in studying retrotransposons is the difficulty in assigning multi-mapped sequencing reads to unique genomic copies/loci (Lanciano and Cristofari, 2020). Our random assessment strategy for multi-mapped sequencing reads yielded consistent conclusions, ruling out computational biases (Fig. EV6F,G). These data reveal that m⁶A marks retrotransposon-derived RNAs and their stage-specific expression during the OET process, suggesting a potential role in the post-transcriptional regulation of both maternal and ZGA retrotransposon-derived RNAs. It is likely that the widespread distribution of m⁶A marks on retrotransposons is largely 'hard-wired' by the genomic architecture (He et al, 2023; Uzonyi et al, 2023; Yang et al, 2022).

Human reproduction is regulated by retrotransposons (Macfarlan et al, 2012; Xiang et al, 2022) and studies on the potential intervention and influence of m⁶A status on retrotransposons in human embryo development is intriguing. Due to the limited availability of human early embryos and ethical considerations, investigating m⁶A status and retrotransposon activity in human early embryos is currently not feasible. Instead, to observe the possible effect of inhibiting m⁶A levels of retrotransposon-derived RNAs, we resorted to use hESCs as a model. We treated hESCs with STM2457, a highly potent and selective catalytic inhibitor of the m⁶A writer METTL3 (Yankova et al, 2021). As expected, at the transcriptome level, STM2457 treatment led to a considerable reduction of uniquely mapped sequencing reads (>50%) (Fig. EV6H), and a substantial loss of ~73% in the number of m⁶A peaks (Fig. EV6I). In the context of retrotransposons, this treatment

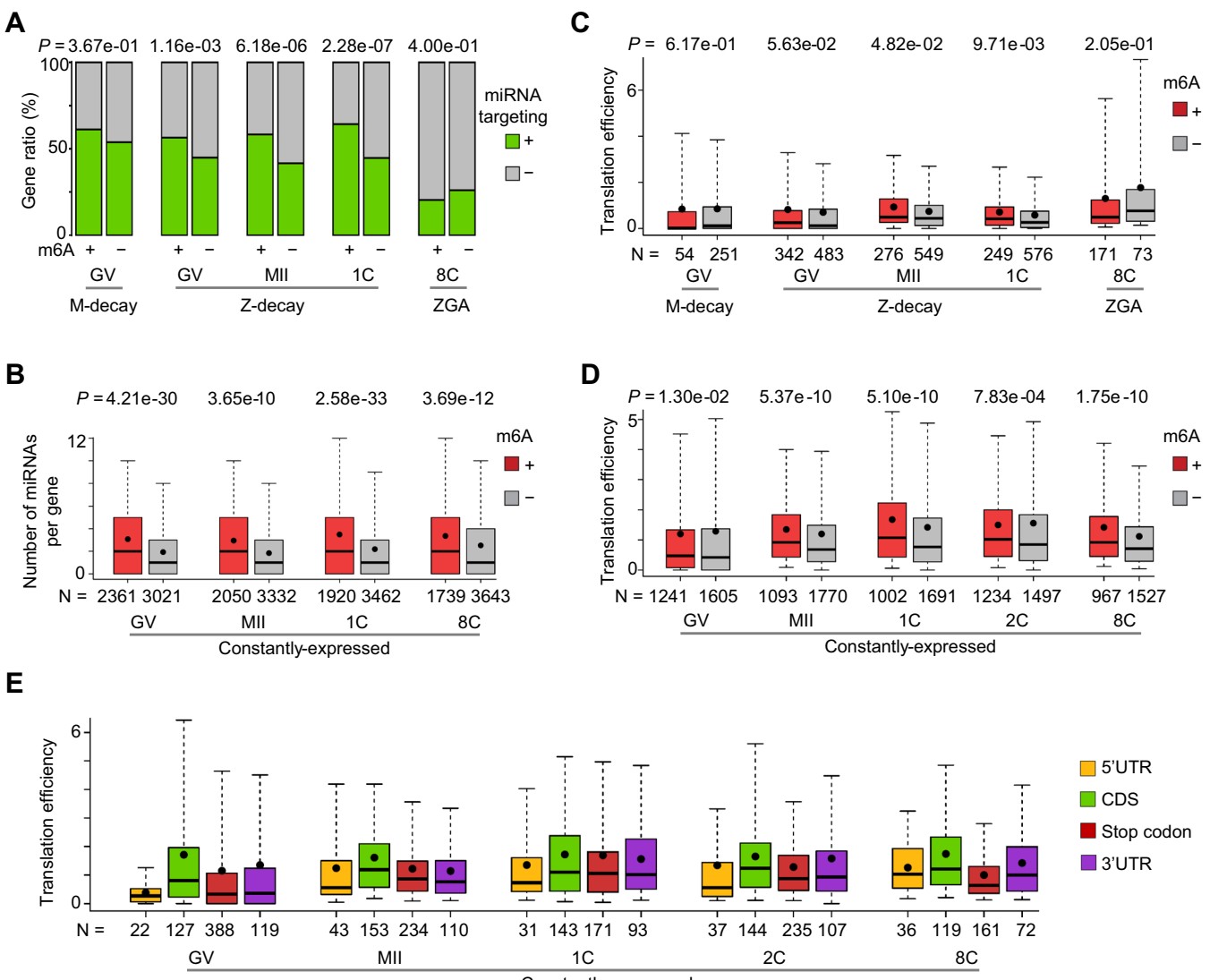

**Figure 4. Association of m⁶A modification with miRNA targeting and translation efficiency.**

(A) Comparison of the proportion of mRNA genes targeted by miRNAs between m⁶A modified and unmodified genes for M-decay, Z-decay and ZGA genes. The *P* values are calculated by the one-sided Fisher's exact test. (B) Comparison of the count of miRNAs targeting m⁶A modified versus unmodified genes for constantly expressed mRNA genes. The *P* values are calculated by the two-sided Wilcoxon rank-sum test. *N* number of genes used in the data analysis. (C, D) Comparison of translation efficiency between m⁶A modified and unmodified genes for M-decay, Z-decay and ZGA mRNA genes (C), as well as for constantly expressed mRNA genes (D). The *P* values were calculated by the two-sided Wilcoxon rank-sum test. *N* number of genes used in the data analysis. (E) Comparison of translation efficiency in constantly expressed mRNA genes with m⁶A across different genic regions. *N* number of genes used in the data analysis. The boxplots in (B–E) were generated with five percentile-based whiskers, arranged from top to bottom as follows: the 95th percentile, the 75th percentile, the median, the 25th percentile, and the 5th percentile; and the mean value indicated by a dot.

resulted in an around 47% reduction in m⁶A peaks associated with retrotransposons (Fig. 5C). The observed variation in reduction rates could be due to differences in transcript structure and the inherent transcriptional regulation of retrotransposons. Retrotransposons, as repetitive elements, may have sequence-specific features that make them less sensitive to m⁶A modification machinery inhibitors. Of note, m⁶A levels on retrotransposons, including those from LTR (e.g., HERVH, HERVK, HERVL, HERV9NC, MLR1B, MSTA, THE1C) and LINE1 (e.g., L1HS, L1PA2, L1PA3), were significantly diminished after STM2457 treatment (Figs. 5D and EV7A–C). Following STM2457 treatment, more than half of the genes involved in transcription and

metabolic pathways lost m⁶A modification, yet their expression levels remained largely unchanged (Fig. EV7D–F). The retrotransposon stage-specific expression patterns during human OET, combining with the analysis of METTL3 inhibitor experiment collectively highlight the critical role of m⁶A in labeling active retrotransposon transcripts.

# Discussion

To the best of our knowledge, no studies to date have explored RNA modifications transcriptome-wide in human oocytes and

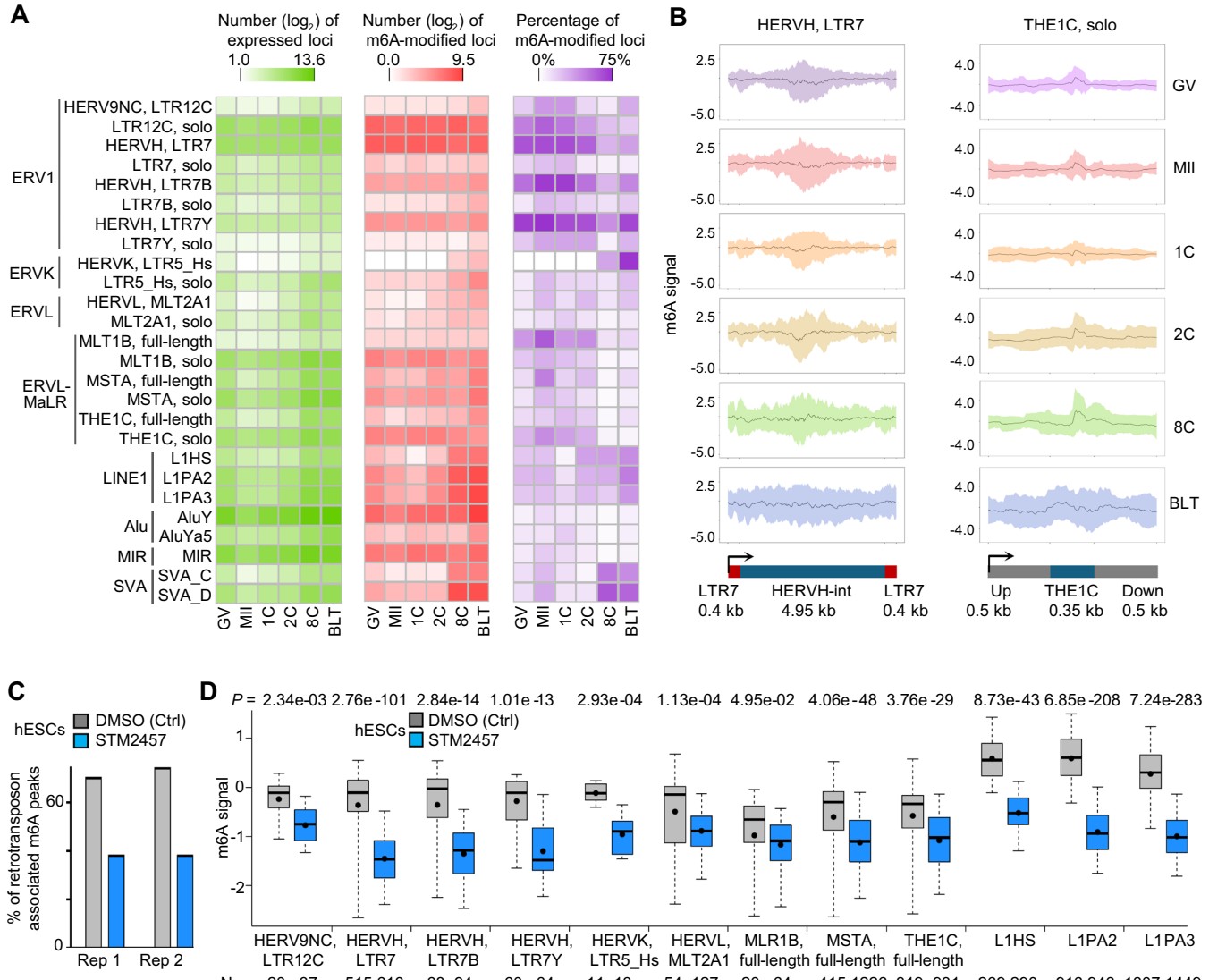

**Figure 5. m⁶A deposition on human retrotransposon-derived RNAs.**

(A) Heatmaps showing the number of expressed loci (left), the number of those with m⁶A among the expressed loci (middle), and the percentage of loci with m⁶A (right) for the representative retrotransposon subfamilies. These results are based on uniquely aligned reads. (B) Distribution of m⁶A signals along the full-length HERVH and solo THE1C sequences (based on uniquely aligned reads from biological replicate 1). The mean (represented by the central black line) and standard deviation (represented by the band around the black line) across all loci were plotted. (C) Comparison of the number of retrotransposon-associated m⁶A peaks between DMSO control (Ctrl) and STM2457 treatment groups in human ESCs. (D) Comparison of the m⁶A signal on representative retrotransposon subfamilies between DMSO and STM2457 groups in hESCs (based on uniquely aligned reads from biological replicate 1). The P values were calculated by the two-sided Wilcoxon rank-sum test. The boxplots were generated with five percentile-based whiskers, arranged from top to bottom as follows: the 95th percentile, the 75th percentile, the median, the 25th percentile, and the 5th percentile; and the mean value indicated by a dot. N number of repeat loci used in the data analysis.

embryos, leaving the roles of RNA modifications in human preimplantation embryos unknown. Here, we first explored this long-standing question of how conserved RNA m⁶A reprogramming is between mouse and human early embryos. We observed the highly dynamic landscape of the m⁶A methylome on both coding and non-coding RNAs (Fig. 6). Recent advances in understanding the mechanism of m⁶A deposition suggest that m⁶A functions akin to a hardware feature that is added by default (He et al, 2023; Uzonyi et al, 2023; Yang et al, 2022). Its specificity is globally regulated by suppressors that prevent m⁶A deposition in

unmethylated transcriptome regions. EJC have been identified as an m⁶A suppressor that protect exon junction–proximal RNA within coding sequences from being methylated. Longer genes tend to have more exons and splicing events, increasing the availability of RRACH motifs in exonic regions. Our findings are consistent with this unified model; indeed, even during the transition from oocyte to embryo, m⁶A-modified mRNAs consistently exhibit longer gene lengths and a greater number of RRACH motifs. We also revealed distinct m⁶A methylome patterns during human major ZGA compared to mice. These differences likely reflect

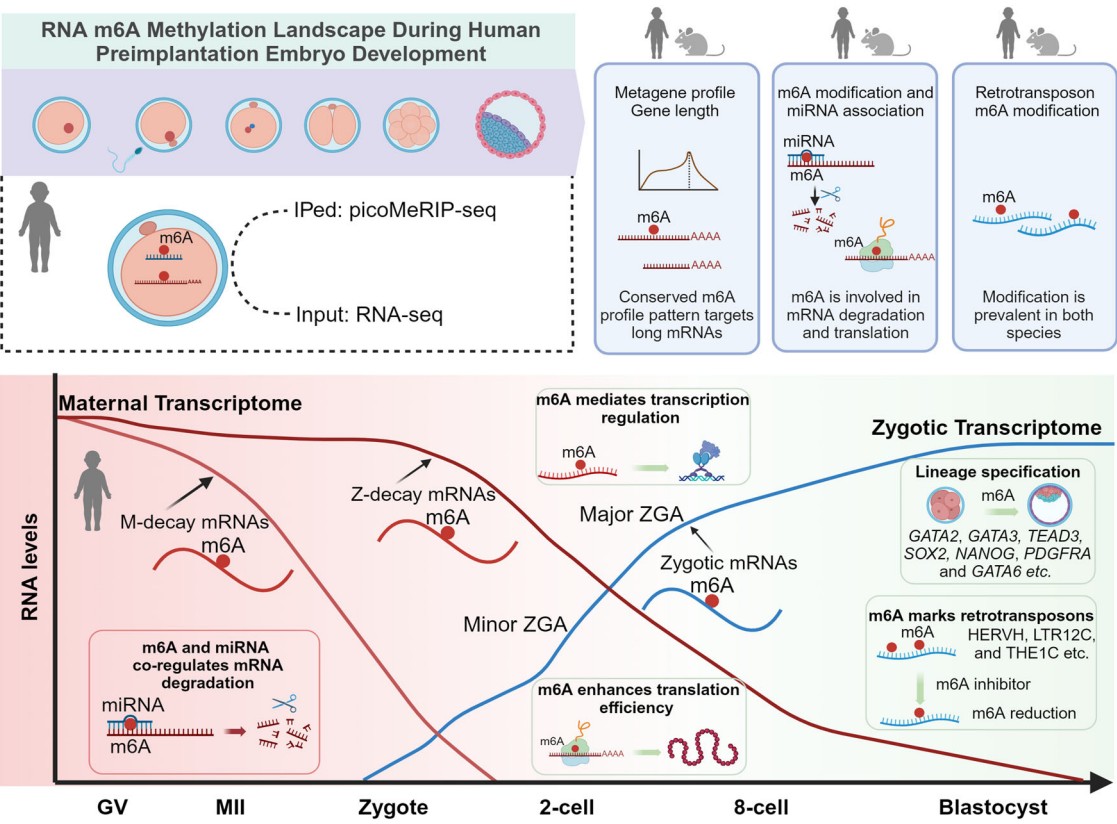

**Figure 6.** Summary of the RNA m⁶A methylome landscape during mammalian OET.

evolutionary adaptations to species-specific needs. We explored the potential functions of m⁶A in the degradation of M-decay and Z-decay maternally expressed genes (Fig. 6). m⁶A-modified transcripts were significantly targeted by miRNAs, pointing to a possible collaborative mechanism in RNA decay. We also observed that m⁶A-modified genes, which were consistently expressed across all developmental stages, exhibited higher translational efficiency (Fig. 6). These observations prompted several important questions for future research, such as the precise mechanisms by which m⁶A promotes RNA decay and translation. We found m⁶A modifications are frequently enriched in stage-specifically expressed retrotransposons, especially within young subfamilies such as HERVH, L1HS, and SVA_D (Fig. 6). It will be interesting to determine whether co-transcriptional incorporation of m⁶A in retrotransposons is related with ZGA. We also evidenced the retrotransposon m⁶A levels was downregulated by METTL3 inhibitor STM2457 in hESCs (Fig. 6). Due to the limited availability of human embryos, we were unable to conduct large-scale single-embryo analyses to explore heterogeneity or experimentally validate these intriguing questions. By utilizing picoMeRIP-seq to explore the RNA m⁶A landscape in human embryos, we aim to inspire further research into RNA modifications and their roles in early human embryonic development.

In summary, our study offers a comprehensive overview of the dynamic m⁶A methylome in human oocytes and embryos, highlighting both conserved and human-specific features compared to mice (Fig. 6). The mechanisms by which m⁶A influences mRNA stability and translation require further elucidation to fully understand the epitranscriptomic regulatory programs governing early mammalian embryonic development. Future research should aim to unravel these intricate processes to provide deeper insights into the role of m⁶A in developmental biology and its implications for human reproduction health.

## Methods

**Reagents and tools table**

| Reagent/resource | Reference or source | Identifier or catalog number |
| --- | --- | --- |
| **Experimental models** | | |
| Tumor samples from patients (*H. sapiens*) | This study | |
| Human embryonic stem cells | ATCC | H1-hESC cell line |
| **Antibodies** | | |
| Anti-m6A polyclonal antibody | Merck Millipore | Cat #: ABE572; Lot #: 3194595 |
| **Chemicals, enzymes and other reagents** | | |
| RNase-free water | Life Technologies | Cat #: 10977035 |

| Reagent/resource | Reference or source | Identifier or catalog number |
|---|---|---|
| Tris-HCl pH 7.5 solution | Life Technologies | Cat #: 15567-027 |
| Tris-HCl pH 8.0 solution | Life Technologies | Cat #: 15568025 |
| NaCl solution | Sigma-Aldrich | Cat #: 71386 |
| NP-40 | Thermofisher Scientific | Cat #: 85124 |
| RiboLock RNase inhibitor | Thermofisher Scientific | Cat #: EO0381 |
| NaCl solution | Sigma-Aldrich | Cat #: 71386 |
| Triton X-100 | Merck Life Sciences | Cat #: T8787-50ML |
| SDS solution | Thermofisher Scientific | Cat #: 11538896 |
| EDTA pH 8 solution | BioWORLD | Cat #: 40520000 |
| EGTA pH 8 solution | BioWORLD | Cat #: 50-255-956 |
| Sodium deoxycholate, 10% solution | BioWORLD | Cat #: 40430018-1 |
| Proteinase K | New England Biolabs | Cat #: P8107S |
| 3 M sodium acetate, pH 5.2 | Thermofisher Scientific | Cat #: R1181 |
| Linear acrylamide | Thermofisher Scientific | Cat #: AM9520 |
| SMART-Seq® Stranded Kit | Takara | Cat #: 634442 |
| STM2457 | MCE | Cat #: HY-134836 |
| Dynabeads™ Protein A for Immunoprecipitation | Thermofisher Scientific | Cat #: 10001D |
| NEBNext rRNA depletion kit | New England Biolabs | Cat #: E6310L |
| Hijack the library quantification kit | Roche | Cat #: kk4824 |
| High Sensitivity D1000 ScreenTape | Agilent | Cat #: 5067-5584 |
| **Software** | | |
| FastQC (v0.11.8) | https://www.bioinformatics.babraham.ac.uk/projects/fastqc/ | |
| Cutadapt (v1.8.1) | EMBnetjournal (Marcel, 2011) | |
| HISAT2 (v2.1.0) | Nat Methods (Kim et al, 2015) | |
| SAMtools (v1.9) | Bioinformatics (Li et al, 2009) | |
| BEDTools (v2.28.0) | Bioinformatics (Quinlan and Hall, 2010) | |
| StringTie (v1.3.5) | Nat Biotechnol (Pertea et al, 2015) | |
| NormExpression (v0.1.0) | Front Genet (Wu et al, 2019) | |
| deepTools (v3.2.0) | Nucleic Acids Res (Ramirez et al, 2014) | |
| MACS2 (v2.1.2) | Genome Biol (Zhang et al, 2008) | |
| MetaPlotR | Bioinformatics (Olarerin-George and Jaffrey, 2017) | |
| Homer (v4.11.1) | Mol Cell (Heinz et al, 2010) | |

| Reagent/resource | Reference or source | Identifier or catalog number |
|---|---|---|
| BioQC (v1.10.0) | BMC Genomics (Zhang et al, 2017) | |
| DAVID (v6.8) | Nat Protoc (Huang et al, 2009) | |
| **Other** | | |
| Illumina NovaSeq sequencing | Illumina | Paired-end mode |

## Ethics statement

To comply with ethical permissions and the European GDPR regulations, human embryo work approval was obtained from regional committees for medical and healthcare research ethics (Regionale komiteer for medisinsk og helsefaglig forskningsetikk, reference ID 2016/1263). hESCs work adheres to the International Society for Stem Cell Research (ISSCR) guidelines. Human oocytes and sperm for the study were sourced from patients undergoing fertility treatment at the Department of Reproductive Medicine, Oslo University Hospital (OUH). These materials were collected only after obtaining voluntary informed consent from the patients and their partners. Patients were given the option of continued storage, disposal, or donation of embryos to this research (RNA modifications during early human embryo development). Patients received no compensation and were offered counseling. They could withdraw their participation at any time until the embryo was used for research. This project did not affect the patients' IVF cycles or normal treatment processes. After embryo evaluation and selection for patient treatment by individual embryologists, only suboptimal quality oocytes and embryos, deemed unsuitable for clinical treatment, were collected for this project. Embryos were not cultured beyond day 5 post-fertilization. Suboptimal oocytes and embryos were collected and transported immediately to the laboratory for processing.

## Human ESC, oocyte and early embryo collection

Patients underwent controlled ovarian stimulation with recombinant or urine-derived follicle stimulating hormone (FSH). Spontaneous ovulation was prevented by either co-administration of GnRH antagonist or suppression with GnRH agonist from the mid-luteal phase of the preceding cycle. Final follicular maturation was induced by administering human chorionic gonadotropin (hCG) upon observation of ≥3 follicles with a diameter of ≥17 mm via ultrasound monitoring. Oocyte retrieval was performed 36 h after hCG administration under transvaginal ultrasound guidance. Retrieved embryos were cultured in Vitrolife G1, G2, or G-TL media under controlled conditions of 6% $CO_2$ and 37 °C, in accordance with standard clinical protocols at the Department of Reproductive Medicine, OUH. Healthy sperm samples were cryopreserved in liquid nitrogen using cryoprotectants. Immature oocytes, identified by the presence of a germinal vesicle (GV), were selected for analysis. Single matured (MII) oocytes from fertile, healthy donors were included for single-cell analysis. These oocytes underwent vitrification and warming using RapidVit and Rapid-Warm oocyte kits (Vitrolife, Sweden). Intra cytoplasmic sperm

injection (ICSI) was employed for fertilization. Fertilization was assessed post-ICSI, and embryo cleavage was monitored every 12 h. Suboptimal grade embryos, deemed unsuitable for cryopreservation, were donated to the project with patient consent. Single human oocytes and embryos were manually sorted into 12 µl of 1× lysis buffer (Takara). The samples were snap-frozen in liquid nitrogen and stored at −80 °C until further analysis.

Human embryonic stem cells (H1-hESC cell line) obtained from ATCC were cultured in feeder-free conditions. Regular mycoplasma testing was conducted, confirming the absence of mycoplasma contamination in all cell lines. For METTL3 inhibition, STM2457 (MCE, HY-134836) was added into the medium at a final concentration of 5 µM for 2 days (Yankova et al, 2021). Cells were sorted into 1× lysis buffer (Takara) in an 8-strip PCR tube using a BD FACSMelody cell sorter (BD Biosciences) following the manufacturer's instructions. Plates were sealed and stored at −80 °C until further processing.

## picoMeRIP-seq experiments

The whole procedures were conducted within a UV decontaminated LAF bench.

### rRNA and DNA depletion

To achieve single-tube rRNA and DNA depletion, we employed the NEBNext rRNA depletion kit (NEB) with slight modifications to the manufacturer's protocol. Initially, RNA/probe master mix (3 µl) was added to a 12-µl sample. Subsequently, the sample underwent a precise temperature ramp from 95 °C to 22 °C, controlled at a rate of −0.1 °C s$^{-1}$, followed by a 5-min incubation period at 22 °C. The RNase H reaction mix (5 µl) was then added to the samples, which were incubated at a precisely controlled temperature of 37 °C for 30 min. Following this, the samples were treated with 30 µl of DNase I digestion mix and incubated at 37 °C for an additional 30 min. The resulting samples were purified using 2.2× volume of RNAClean XP beads, washed twice with 80% freshly prepared ethanol, and finally eluted with 78 µl of nuclease-free water. To prevent RNA degradation, 2 µl RiboLock RNase inhibitor (40 U µl$^{-1}$) was added to each sample, ensuring a final sample volume of 80 µl.

### RNA fragmentation by sonication

Sonication of the samples was carried out using a UP100H Ultrasonic Processor (Hielscher) equipped with a 2-mm probe. The samples underwent $2 \times 30$ s sonication cycles, precisely alternating between 30 s of sonication and 30 s on ice for each cycle. After sonication, 20% RNA fractions from several oocytes or embryos were pooled as input controls for each developmental stage. Add appropriate nuclease-free water to ensure a final volume of 80 µl. After that, 20 µl 5× IP buffer was added to the samples to achieve a final volume of 100 µl for subsequent processing steps.

### Antibody–bead incubation

Prior to use, Dynabeads (Invitrogen) underwent washing to ensure purity. Specifically, 20 µl of beads was washed twice with 1× IP buffer, with each washing step involving vortexing, rapid centrifugation on a MiniGalaxy, and placement on a magnetic rack to remove the supernatant. Following this, the antibody solution was prepared by diluting 4 µl of Millipore anti-m⁶A (Cat num: ABE572; Lot num: 3194595) antibody in 16 µl of 5× IP buffer

(50 mM Tris-HCl (pH 7.5), 750 mM NaCl, 0.5% (vol/vol) NP-40, 5 U/µl RiboLock) and 60 µl of nuclease-free water. The antibody-containing solution (80 µl) was added to the washed beads, and the antibody–bead mixture was incubated overnight with head-over-tail rotation on a HulaMixer at 4 °C (40 r.p.m.).

### Immunoprecipitation (IP) and washes

The prepared antibody-coated beads were captured on the tube wall using a magnetic rack, and the supernatant from the antibody–bead incubation was carefully discarded. The antibody-coated beads then underwent washing to remove unbound antibodies that could otherwise compete for binding to the epitope. The beads were washed twice with 200 µl of 1× IP buffer, with each wash involving vortexing (four times for 5 s each) to ensure thorough washing. After vortexing during the second wash, the antibody-coated beads were transferred to 0.2-ml PCR tubes, and a volume of 10 µl of homogenously antibody-coated bead solution was transferred to each PCR tube. The tubes were rapidly centrifuged on a MiniGalaxy and placed on a magnetic rack for at least 2 min or until the solution became clear. Following removal of the supernatant, 100 µl of the sonicated RNA sample was added to each antibody–bead-containing tube, and the samples were incubated with head-over-tail rotation on a HulaMixer at maintained 4 °C for 2 h (40 r.p.m.). Subsequently, the tubes were swiftly centrifuged on a MiniGalaxy and placed on a magnetic rack. The supernatant was meticulously removed, and the RNA–antibody–bead complexes underwent a series of washes to eliminate non-specific binding. The RNA–antibody–bead complexes were washed four times in the following solutions, quickly spun and placed in a magnetic rack in between washes: washed once with ice-cold medium-stringency RIPA buffer (10 mM Tris-HCl (pH 8.0), 300 mM NaCl, 1 mM EDTA, 0.5 mM EGTA, 1% (vol/vol) Triton X-100, 0.2% (vol/vol) SDS and 0.1% (vol/vol) sodium deoxycholate), washed twice with ice-cold high-stringency RIPA buffer (10 mM Tris-HCl (pH 8.0), 350 mM NaCl, 1 mM EDTA, 0.5 mM EGTA, 1% (vol/vol) Triton X-100, 0.23% (vol/vol) SDS and 0.1% (vol/vol) sodium deoxycholate) and washed once with ice-cold medium-stringency RIPA buffer. Following the four wash steps, the tubes were centrifuged and placed on a magnetic rack, and the supernatant was carefully discarded. The RNA–antibody–bead complexes were then resuspended in 100 µl of 1× IP buffer and incubated for 5 min. Following this, the samples were swiftly centrifuged and placed on a magnetic rack, and the supernatant was removed. The RNA–antibody–bead complexes were resuspended in 147.9 µl of elution buffer (5 mM Tris-HCl (pH 7.5), 1 mM EDTA, 0.05% (vol/vol) SDS, and 1 U µl$^{-1}$ RiboLock RNase inhibitor). Proteinase K (2.1 µl; NEB) was added to each tube, and the tubes were then incubated on a Thermomixer at 1200 r.p.m. and 55 °C for 1.5 h. Following incubation, the tubes were briefly centrifuged and incubated further on a Thermomixer at 80 °C for 20 min to inactivate the Proteinase K. Subsequently, the samples were placed on a magnetic rack for 2–3 min, and the supernatant containing the m⁶A-immunoprecipitated RNA was carefully transferred to a new 1.5-ml low-binding tube. The remaining beads were resuspended again in 147.9 µl of elution buffer, and 2.1 µl of Proteinase K was added. The samples were immediately placed on a Thermomixer at 1200 r.p.m. and 55 °C for 5 min, followed by inactivation of Proteinase K on a Thermomixer at 80 °C for 20 min. The tubes were then placed back on a magnetic rack for 2–3 min, and the supernatant was collected and pooled

with the first supernatant in the same 1.5-ml low-binding tube to recover as much of the m⁶A-immunoprecipitated RNA as possible, resulting in a total volume of ~300 µl.

### Ethanol precipitation

For both input and immunoprecipitated RNA samples, nuclease-free water was added to each tube to achieve a final volume of 400 µl. Subsequently, 40 µl of 3 M sodium acetate (pH 5.2; Thermo Fisher Scientific) and 10 µl of linear acrylamide 5 mg µl⁻¹ (Thermo Fisher Scientific) were added to each sample, followed by 1000 µl of ice-cold 100% ethanol. The samples were vigorously vortexed without centrifugation or spinning and immediately placed at –80 °C for at least 2 h or overnight until completely frozen. Once the samples were recovered from –80 °C, they were allowed to briefly thaw on ice, ensuring visual confirmation that all samples had thawed before commencing centrifugation. The samples were then centrifuged at 20,000× g at 4 °C for 15 min, and the supernatant was removed without disturbing the visible pellet. The pellet was carefully washed twice with 1 ml of ice-cold 75% ethanol. For each wash, 75% ethanol was added, centrifugation was then repeated with 10 min at 4 °C. Following the last wash, as much as possible of the supernatant was removed, and the tube lid was left open until all ethanol had evaporated, resulting in a dried pellet. The dried pellet was resuspended in 7 µl of nuclease-free water for library preparation.

### Library preparation and sequencing

With modifications to the manufacturer's protocol, as described below, the SMART-Seq stranded kit (Takara, 634442) was used to construct sequencing libraries. For the fragmented input or immunoprecipitated RNA, the protocol was performed without the fragmentation step. After the first PCR amplification and following AMPure bead purification, the beads were resuspended by adding 46.5 µl of nuclease-free water, and the ribosomal cDNA depletion protocol was skipped. The samples were incubated at room temperature for 5 min for rehydration, and 46 µl of supernatant was recovered from each sample. Subsequently, the protocol was followed until completion. The libraries were assessed for quantity using KAPA library quantification kits (Roche), and size distribution was evaluated using TapeStation D1000 ScreenTape (Agilent Technologies). These assessments provided a good estimation of pooling at equimolar ratios. The pooled libraries were sequenced on a NovaSeq system (Illumina).

## picoMeRIP-seq data processing and m⁶A peak calling

### Quality control, alignment, and processing of sequencing reads

The quality of the raw sequencing reads for both Input and IP samples was evaluated using FastQC (v0.11.8) (https://www.bioinformatics.babraham.ac.uk/projects/fastqc/), and sequencing adapters were trimmed using Cutadapt (v1.8.1) (Marcel, 2011) with settings "-m 20 --max-n 0.01 --trim-n". The trimmed reads were aligned to the human reference genome (hg38) using HISAT2 (v2.1.0) (Kim et al, 2015) with the parameter "−5 8 --no-mixed --no-discordant". We retained only those read pairs aligning uniquely to one genomic location, as indicated by HISAT2. PCR duplicates were removed using SAMtools (v1.9) (Li et al, 2009) fixmate & markdup, and the read pairs aligning to ribosomal RNAs (per GENCODE v39 annotation) were filtered out using BEDTools (v2.28.0) (Quinlan and Hall, 2010) intersect function.

### Gene and transcript quantification from the Input samples

Based on GENCODE (v39) gene annotation, the expression abundance (TPM, transcript per million) of genes and transcript/isoforms were estimated by StringTie (v1.3.5) (Pertea et al, 2015) with the parameter "-e -A". Further, the normalization of expression abundances among samples was performed by the R package NormExpression (v0.1.0) (Wu et al, 2019) with the normalization factor (method = "DESeq").

### Definition of m⁶A signal

The genome coverage bigWig files with a resolution of 10 bp (i.e., bin size = 10 bp) and normalization according to RPKM (reads per kilobase per million reads) were generated by deepTools (v3.2.0) (Ramirez et al, 2014) bamCoverage with the parameter "-bs 10 --normalizeUsing RPKM". The visualization of read density along exonic regions of genes was based on these bigWig files. For each gene, the exonic regions of the transcript/isoform with highest expression value was used to visualize the read density. The transcripts/isoforms used for genome browser visualization are ENST00000687138.1 (GDF9), ENST00000638280.2 (LEUTX), ENST00000263620.8 (ARID3A), ENST00000429829.6 (XIST), ENST00000649650.1 (ZDBF2), ENST00000513960.5 (VCAN), ENST00000345382.7 (BNC1), and ENST00000201031.3 (TFAP2C).

For each bin (size = 10 bp), the m⁶A signal was calculated as the log2 transformed ratio of (the IP sample's RPKM value + 1) over (the Input sample's RPKM value + 1).

### PCA, Pearson correlation, and hierarchical clustering analyses

By deepTools multiBigwigSummary ("bins" mode and window size = 1 kb) & plotPCA & plotCorrelation, we performed principal component analysis (PCA), and calculated Pearson correlation coefficients ($r$) along with hierarchical clustering of the samples. In the hierarchical clustering, the dissimilarity distance between any two samples was defined as $1.0 − r$.

### m⁶A peak calling

MACS2 (v2.1.2) (Zhang et al, 2008) callpeak with the parameter "-g 421727543 --keep-dup all -B --nomodel --call-summits" was used identify m⁶A peaks ($q$ value < 0.05), with the IP sample and the corresponding Input sample.

## Genomic annotation, metagene profiling, and motif analysis of m⁶A peaks

Peak annotation was carried out using BEDTools based on the GENCODE (v39) annotation library. To avoid assigning a peak to multiple genomic features, the hierarchy of genomic features prioritized as follows: stop codon (ranging from 200 bp upstream to 200 bp downstream of the annotated stop codon in GENCODE), 3' UTR, 5′ UTR, CDS (coding sequencing), exon, intron, and intergenic. The enrichment score at each genomic region was calculated as the log2 ratio of the observed over expected peak numbers. The expected number of peaks is calculated based on the length proportion of different genomic regions (i.e., 5'UTR, CDS, 3'UTR, stop codon, exon, intron, intergenic) relative to the total length of reference genome hg38.

Metagene profiles of m⁶A peak summits along protein coding genes were generated using MetaPlotR (Olarerin-George and Jaffrey, 2017). For each gene, only the concatenated exonic regions

of the transcript/isoform with the highest expression value were used for plotting metagene profiles.

We searched for consensus m⁶A motifs using the peak regions as detected by MACS2 and the regions (within a 100 bp, 150 bp, 200 bp, 300 bp, and 400 bp windows centered on peak summits as detected by MACS2), and by Homer (v4.11.1) (Heinz et al, 2010) findMotifsGenome.pl with the parameter "-rna -len 5,6,7,8".

## Definition of m⁶A gene

We defined a gene as m⁶A+ if ≥1 of its transcript/isoform intersected with ≥1 m⁶A peak; otherwise, the gene was considered as m⁶A−. Considering that we had two IP samples for each developmental stage, we opted for a union approach: considering a gene as m⁶A+ if it was m⁶A+ in any of two IP samples.

## Comparison between human and mouse samples

Homologous genes between human and mouse were obtained from the Mouse Genome Informatics (MGI) database (https://www.informatics.jax.org/homology.shtml), on August 17, 2023, using the download link "https://www.informatics.jax.org/downloads/reports/HOM_MouseHumanSequence.rpt". We omitted any gene from the analysis that presented with >1 homologous counterpart between human and mouse.

For mouse samples (oocytes at the GV and MII stages, and embryos at the zygote/1C, 2-cell/2C and blastocyst/BLT stages), we derived their expression abundance and m⁶A profiles from our previous study (Wang et al, 2023).

Defining co-expressed and species-specifically expressed genes: within a given developmental stage that was comparable between human and mouse, a gene having an expression level of TPM ≥ 50 in both species was classified as a co-expressed genes; a gene with TPM ≥ 50 in human but TPM ≤ 10 in mouse was denoted as a human-specifically expressed gene; a gene with TPM ≥ 50 in mouse but TPM ≤ 10 in human was denoted as a mouse-specifically expressed gene.

## Definition of M-decay, Z-decay, ZGA, and constantly expressed genes

M-decay gene: (1) TPM was >10 in GV oocyte; (2) TPM in GV oocyte was greater than in MII oocyte; (3) the expression fold change of GV oocyte versus 8-cell was >2; (4) the expression fold change of GV versus zygote was >2; and (5) the expression fold change of GV versus MII was <2.

Z-decay gene: (1) the TPM was >10 in GV oocyte; (2) the expression fold change of zygote versus GV was <1.5; (3) the expression fold change of GV versus 8-cell was >2; (4) the expression fold change of GV versus zygote was <2; and (5) the expression fold change of zygote versus 8-cell was >2.

MD genes were the combination of M-decay and Z-decay genes.

ZGA gene: (1) TPM < 1 in both GV and MII oocytes; (2) TPM > 10 at the 8-cell stage; and (3) TPM at the 8-cell is greater than at the zygote stage.

Constantly expressed genes: Based on gene expression values across all Input samples, we first calculated the entropy specificity by the R package BioQC (v1.10.0) (Zhang et al, 2017). A gene with <0.5 entropy specificity scores and with TPM ≥ 10 across all samples was defined as constantly expressed one.

## Annotation of transcription factor, transcription cofactor and imprinted genes

The annotation of human transcription factors and transcription cofactors was downloaded from the database AnimalTFDB4 (Hu et al, 2019) on August 30, 2023. Human imprinted genes were obtained from Geneimprint database (https://www.geneimprint.com/site/genes-by-species) on August 30, 2023.

## miRNA targeting analyses

The miRNA expression data (read counts) from four developmental stages (GV, MII, zygote/1C and 8-cell/8C) were obtained from the Dataset EV2 of a previous study (Paloviita et al, 2021). We further normalized the expression of miRNA to reads per million (RPM) within each sample. Given that there were multiple replicates for each developmental stage in the study (Paloviita et al, 2021), we computed the mean RPM value for each miRNA by averaging across all replicates of a given developmental stage.

To reduce the likelihood of false-positive miRNA target predictions, we focused on those miRNAs that were from conserved miRNA families and are highly expressed (RPM ≥ 10). Only the conserved gene targets of the conserved miRNAs were considered based on the miRNA and target gene annotations from TargetScanHuman (v8.0) (Agarwal et al, 2015). A gene can be targeted by multiple different miRNAs. A gene was considered miRNA-targeted at a particular stage if it was predicted to be a target of ≥1 miRNA; and we then counted the number of regulated miRNAs and the weighted number (i.e., the sum of log10 (RPM) of all targeted miRNAs) of regulated miRNAs.

Of note, only the protein coding genes (mRNAs) were used for miRNA analysis.

## Analyses of human translatome data

We obtained translatome (Ribo-lite, single-end) and corresponding transcriptome (mRNA-seq, paired-end) data from a previous study (Zou et al, 2022), available at the NCBI GEO under accession number GSE197265, with two replicates for each developmental stage. The quality of sequencing reads was evaluated by FastQC, and the sequencing adapters were trimmed using Cutadapt with the parameter "cutadapt -a AGATCGGAAGAGCACACGTCTGAA CTCCAGTCAC -u 3 -m 18 --max-n 0.01 --trim-n <input.file> | cutadapt -a AAAAAAAAAACAAAAAAAAAAAAAAAAAAAAA AAAAAAAAAA -m 18 -o <output.file > -" for Riob-lite data, and with the parameter "cutadapt -a AAGCAGTGGTATCAACGCAG AGTACTTTTTTTTTTTTTTTTTTTTTTTTTTTTTTTTTTTTT-TTTTTT -a CTGTCTCTTATACACATCTCCGAGCCCACGAGA C -A CTGTCTCTTATACACATCTGACGCTGCCGACGA -A AA GCAGTGGTATCAACGCAGAGTACTTTTTTTTTTTTTTTTTT TTTTTTTTTTTTTTTTTTTTTTTTTT -G AAGCAGTGGTAT CAACGCAGAGTAC -g AAGCAGTGGTATCAACGCAGAGTAC -m 20 --max-n 0.01 --trim-n" for mRNA-seq data. Using the same procedures as picoMeRIP-seq data (see above), both Ribo-lite and mRNA-seq data underwent alignment (HISAT2 with the default parameter for Ribo-lite, and with the parameter "−5 8 --no-mixed --no-discordant" for mRNA-seq), extraction of uniquely aligned read pairs, removal of PCR duplicates and rRNA-derived read pairs, and estimation of gene abundance (FPKM, Fragments Per

Kilobase of transcript per Million mapped reads). We averaged the FPKM values over two replicates for each developmental stage, and focused on highly expressed mRNA genes (FPKM ≥ 10, estimated by mRNA-seq data) to calculate translation efficiency as the fold change of (Ribo-lite's FPKM + 1) versus (mRNA-seq's FPKM + 1).

Of note, only the protein coding genes (mRNAs) were used for translation analysis.

### Retrotransposon analyses

#### Retrotransposon annotation

For LINE, SINE and SVA retrotransposons, we extracted the annotation using UCSC Table Browser with the setting "clade = Mammal, genome = Human, assembly = GRCh38/hg38, group = Repeats, track = RepeatMasker, table=rmsk" on March 8, 2020. For LTR retrotransposon, we downloaded the annotation from HERVd database (version 3) (Paces et al, 2004) on January 2, 2024. To minimize the effect of highly-fragmented/truncated retrotransposon copies/loci which lack transcription and transposition activities (Cordaux and Batzer, 2009), (1) for LINE, SINE, SVA and solo LTR, we imposed a completeness threshold whereby only genomic loci/copies with ≥90% completeness were included in the analyses; completeness being defined by the ratio of the annotated retrotransposon sequence length in the human reference genome to that of the transposable element reference sequence's full length. Further, for the full-length ERV but not for solo LTR, only loci/copies exhibiting the complete architecture (comprising 5'LTR, internal sequence and 3'LTR) were considered. To avoid the effects of the expression of regular genes on retrotransposon analyses, we excluded the retrotransposon loci/copies that intersected with exonic regions of regular genes (annotated by GENCODE v39).

#### Expression and m⁶A enrichment of retrotransposons

For each retrotransposon locus/copy defined above, we took into account two metrics: expression value and m⁶A signal value. The expression value was computed as the maximum RPKM observed across all bins (size =10 bp) overlapping the locus, using Input sample. The m⁶A signal value was calculated as the maximum m⁶A signal value across all bins (size =10 bp) overlapping the locus (see "Definition of m⁶A signal" under the section "picoMeRIP-seq data processing and m⁶A peak calling"). For each developmental stage, the mean values (expression or m⁶A signal) across two biological replicates were assigned to this locus. A locus/copy was deemed "expressed" if its expression value was >0; and it was considered as "m⁶A modified" if its m⁶A signal value was >0.

### Visualization of boxplots

All boxplots in this study (except Fig. EV1J) were generated with five percentile-based whiskers, arranged from top to bottom as follows: the 95th percentile, the 75th percentile, the median, the 25th percentile, and the 5th percentile. Additionally, each boxplot includes a single point/dot representing the mean value. The boxplot of Fig. EV1J was directly downloaded from the database EmAtlas, with arranged from top to bottom as follows: the maximal value, the 75th percentile, the median, the 25th percentile, and the minimal value.

### GO analysis

We performed GO analysis with the online tool DAVID (v6.8) (Huang et al, 2009) using all human and mouse genes as background, respectively.

## Data availability

Both raw sequencing data and processed data for hESCs, and processed data for human oocytes and embryos generated in this study are available in GEO under accession number GSE268645. The public m⁶A-seq data for hESCs were obtained from the study (Batista et al, 2014) with the SRA accession numbers SRR1035221, SRR1035222, SRR1035223 and SRR1035224. The expression values of miRNAs at the human GV, MII, 1C and 8C stages were obtained from the Dataset EV2 of the study (Paloviita et al, 2021). The translatome (Ribo-lite) and the corresponding transcriptome (mRNA-seq) data were obtained from the study (Zou et al, 2022) with the GEO accession number GSE197265.

The source data of this paper are collected in the following database record: biostudies:S-SCDT-10_1038-S44318-025-00474-5.

## Peer review information

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

## Acknowledgements

This work was supported by South-Eastern Norway Regional Health Authority, Grant 2018086 (to AK); Research Council of Norway, FRIPRO Researcher Project 275286 (to AK), and partially by the Research Council of Norway through its Centres of Excellence scheme, project number 332713, CRESCO; National Institutes of Health grants R01HG011469 and R01GM136886 (to KFA and YW); Institutional fund from the Gilbert S Omenn Department of Computational Medicine and Bioinformatics, University of Michigan (to KFA and YW); South-Eastern Norway Regional Health Authority Early Career Grant 2016058 and Grants 2018063; 2023098 (to JAD); Research Council of Norway, FRIPRO Grant 289467 (to JAD); Oslo University Hospital Strategic Research Environment Grant (to JAD and GDG). The sequencing service was provided by the Norwegian Sequencing Centre (www.sequencing.uio.no), a national technology platform hosted by the University of Oslo and supported by the "Functional Genomics" and "Infrastructure" programs of the Research Council of Norway and the Southeastern Regional Health Authorities".

## Author contributions

**Yanjiao Li**: Conceptualization; Data curation; Formal analysis; Validation; Investigation; Visualization; Methodology; Writing—original draft; Writing—review and editing. **Yunhao Wang**: Conceptualization; Data curation; Software; Formal analysis; Validation; Investigation; Visualization; Methodology; Writing—original draft; Writing—review and editing. **Aylin Cengiz**: Resources; Investigation; Writing—review and editing. **Kang-Xuan Jin**: Resources; Investigation; Methodology. **Blanca Corral Castroviejo**: Resources. **Xiaolin Lin**: Resources. **Marie Indahl**: Resources. **Rujuan Zuo**: Resources. **Trine Skuland**: Writing—review and editing. **Madeleine Fosslie**: Writing—review and editing. **Maria Biba**: Resources; Supervision. **Xuechen Wu**: Resources. **Peter Fedorcsak**: Resources; Supervision. **Magnar Bjørås**: Resources; Supervision. **Adam Filipczyk**: Resources. **John Arne Dahl**: Methodology; Writing—review and editing. **Gareth D Greggains**: Resources; Supervision; Writing—review and editing. **Kin Fai Au**: Conceptualization; Supervision; Funding acquisition; Writing—review and editing. **Arne Klungland**: Conceptualization; Supervision; Funding acquisition; Project administration; Writing—review and editing.

Source data underlying figure panels in this paper may have individual authorship assigned. Where available, figure panel/source data authorship is listed in the following database record: biostudies:S-SCDT-10_1038-S44318-025-00474-5.

## Disclosure and competing interests statement

The authors declare no competing interests.

# Expanded View Figures

**Figure EV1. Performance evaluation of picoMeRIP-seq in human samples using hESCs and m⁶A profiling in human oocytes and preimplantation embryos; along with the RNA expression levels of m⁶A-related regulatory factors.**

(A) Illustration of picoMeRIP-seq procedure performed on human oocytes and early embryos. The figure was created in BioRender. (B) Genome browser snapshots of two genes with m⁶A enrichment in hESCs. (C) Transcriptome-wide correlation analyses of picoMeRIP-seq experiments (IP samples) among samples from 1000, 100 and 10 cells in hESCs. (D) Heatmap showing the fraction of m⁶A-modified genes (identified in samples of 10 cells) over those identified in samples of 1000 and 100 cells, as well as over those from a previous study in hESCs (Batista et al, 2014). (E) Metagene profiles showing the enrichment of m⁶A peaks along protein-coding genes in hESCs. (F) Consensus motifs identified within m⁶A peaks in hESCs. (G) Principal component analysis (PCA) of transcriptome-wide m⁶A signals. Multiple biological replicates for each stage were generated. (H) Number of m⁶A peaks. For each stage, the averaged (mean) number across biological replicates 1 and 2 are shown. (I) Fraction of m⁶A+ genes under different expression cutoffs. (J) Boxplot showing mRNA levels of m⁶A associated proteins in human oocytes and early embryos. The data were downloaded from a previously published database EmAtlas (Zheng et al, 2023). The boxplots are arranged from top to bottom as follows: the maximal value, the 75th percentile, the median, the 25th percentile, and the minimal value. TB trophoblast.

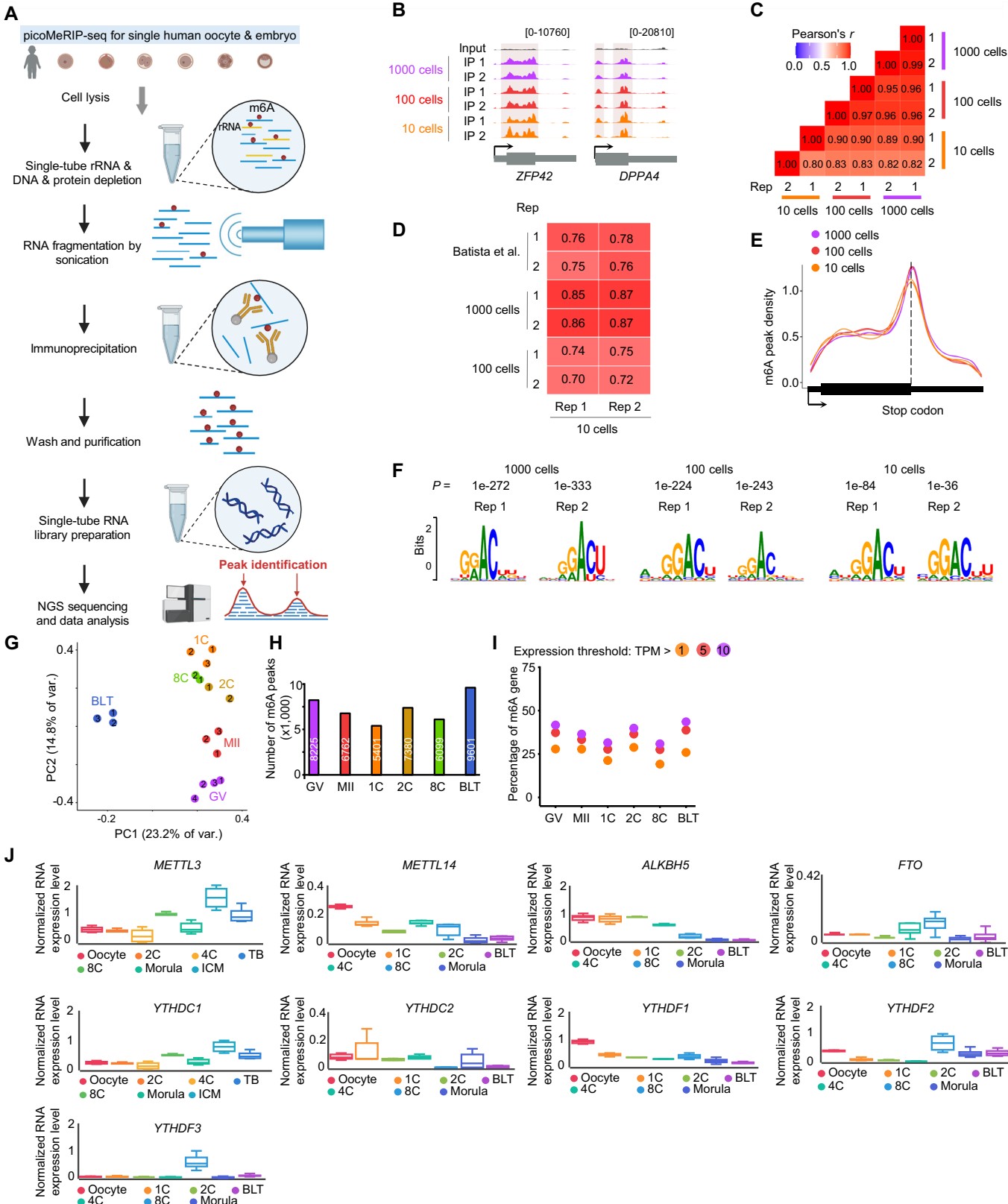

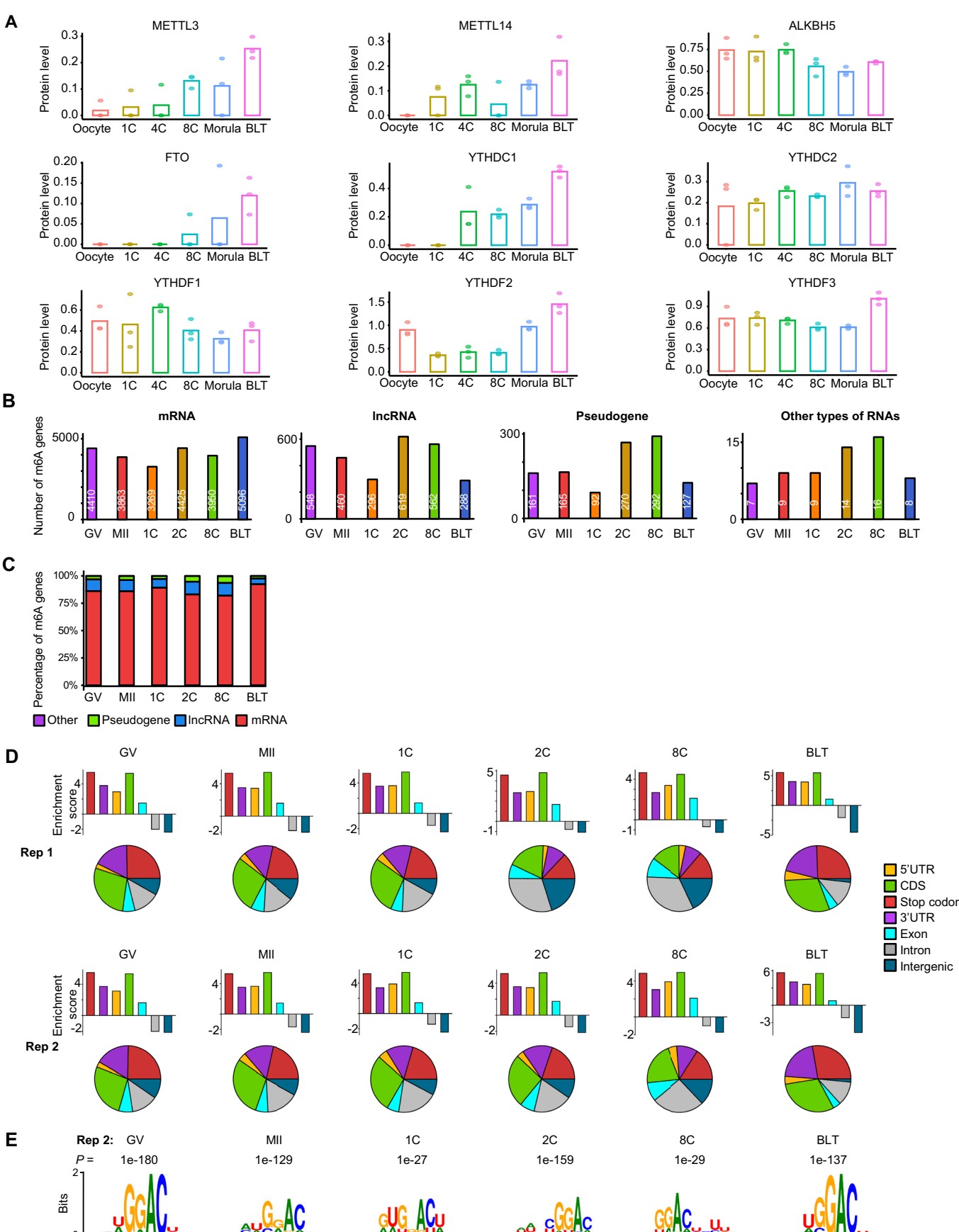

◀ **Figure EV2. The distribution of m⁶A peaks across different developmental stages; along with the protein expression levels of m⁶A-related regulatory factors.**

(A) Bar charts showing protein levels of m⁶A associated proteins in human oocytes and early embryos. The data were downloaded from a previously published database (Zhu et al, 2025). The mean values were shown as the bar. (B) Number of m⁶A+ genes for different types of genes. (C) Fraction of m⁶A+ genes among different types of RNAs. (D) Top, pie charts show genomic annotation of m⁶A peaks. Bottom, bar plots show the peak enrichment score, which was calculated as the $\log_2$ ratio of the observed over expected peak numbers. (E) Consensus motifs identified within m⁶A peaks from biological replicate 2.

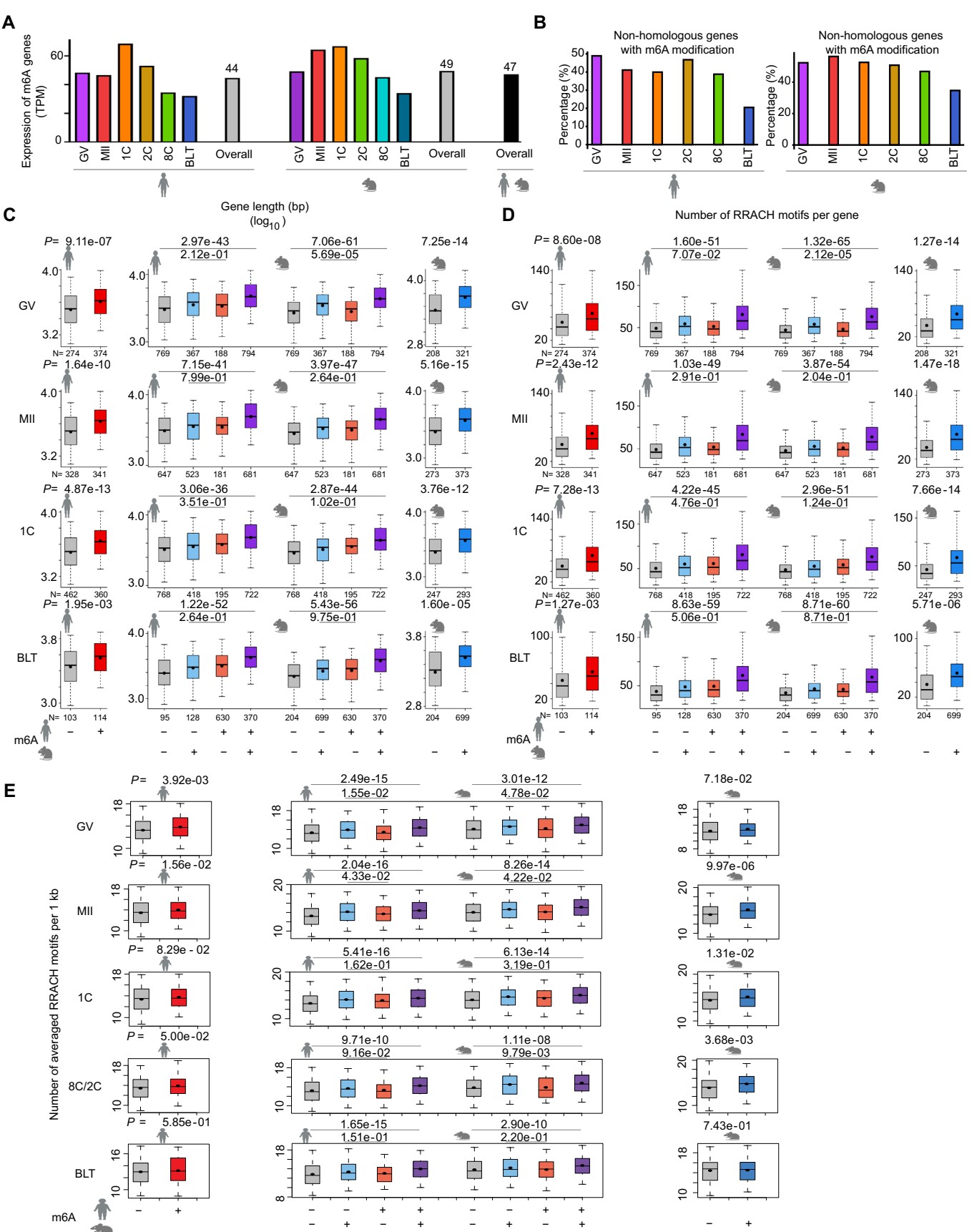

◀ **Figure EV3. Comparison of gene length and RRACH motif between m⁶A modified and unmodified genes in human and mouse.**

(A) Statistics of expression level of m⁶A+ genes in human and mouse data. TPM, transcripts per million. (B) The ratios of non-homologous genes (TPM ≥ 50) with m⁶A modification in human and mouse. (C, D) Comparison of gene lengths (C) and RRACH counts (D) between m⁶A modified and unmodified genes for human- (left) and mouse- (right) specifically expressed genes, as well as human-mouse co-expressed genes (middle). The *P* values were calculated by the two-sided Wilcoxon rank-sum test. *N* number of genes used in the data analysis. (E) The average RRACH motif density after normalized per 1 kb length between m⁶A modified and unmodified genes for human- (left) and mouse- (right) specifically expressed genes, as well as human-mouse co-expressed genes (middle). The *P* values were calculated by the two-sided Wilcoxon rank-sum test. *N* number of genes used in the data analysis. The boxplots in (C–E) were generated with five percentile-based whiskers, arranged from top to bottom as follows: the 95th percentile, the 75th percentile, the median, the 25th percentile, and the 5th percentile; and the mean value indicated by a dot.

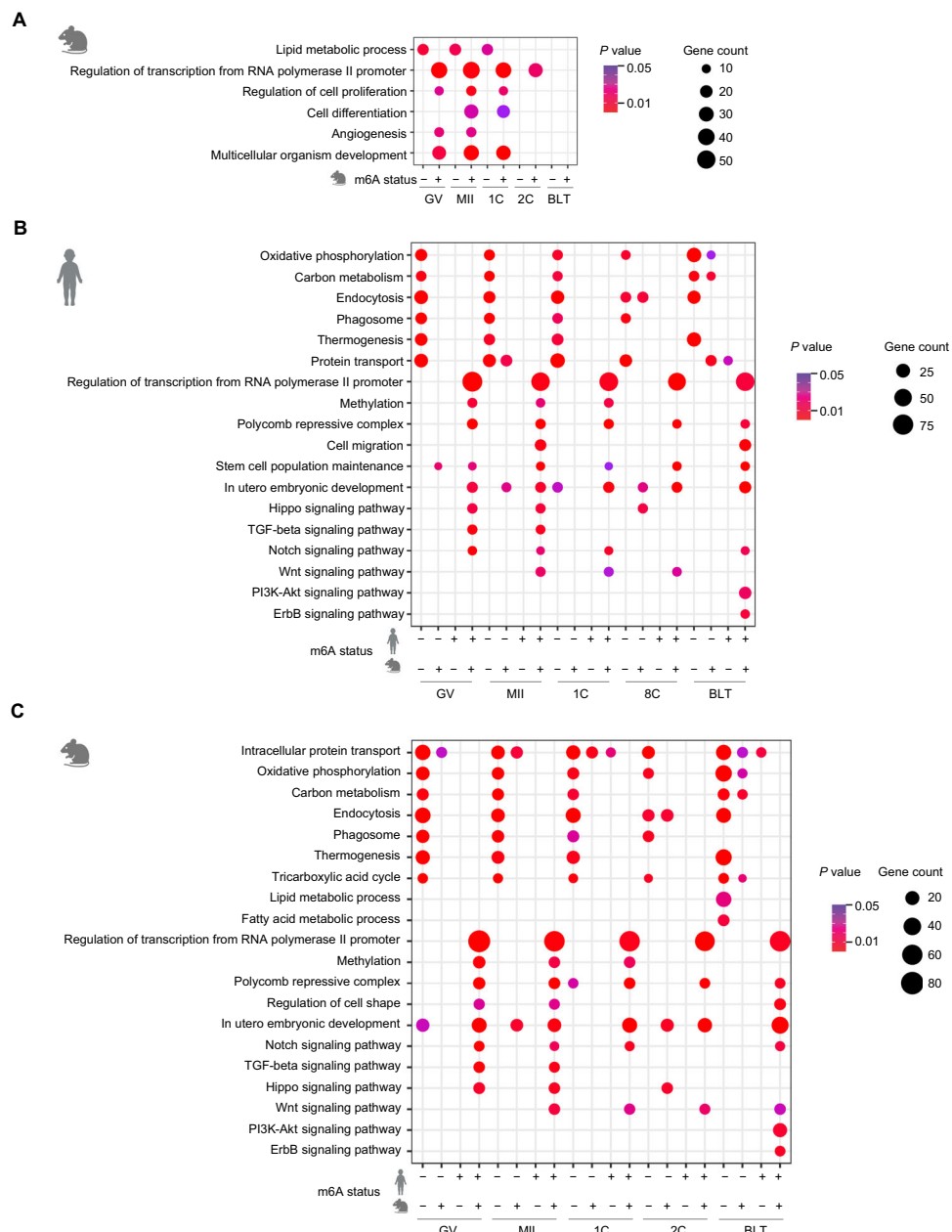

**Figure EV4. GO analysis of m⁶A modified and unmodified genes for mouse-specifically expressed genes and human-mouse co-expressed genes.**

(A–C) GO analysis for m⁶A + /− genes for mouse specifically expressed genes (A), for human genes that were co-expressed in mouse (B), and for mouse genes that were co-expressed in human (C). Fisher's exact test was used to calculate the one-sided $P$ values.

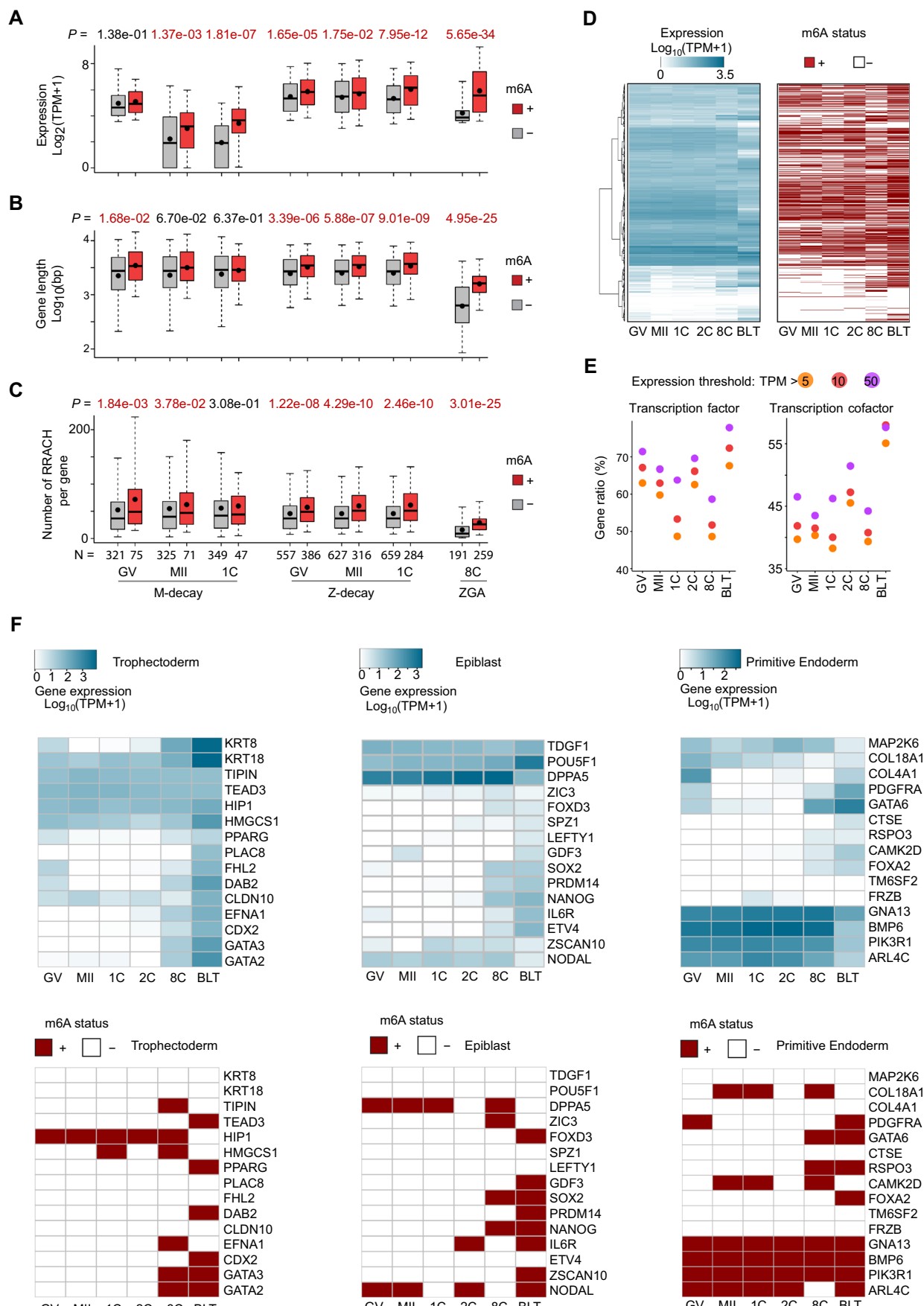

◄ **Figure EV5. Comparison of m⁶A-modified and unmodified genes in terms of M-decay, Z-decay, and ZGA categories; along with the expression levels and m⁶A modification status of transcription factors and cofactors.**

(A–C) Comparison of expression levels (A), gene lengths (B), and RRACH counts (C) between m⁶A modified and unmodified genes for M-decay, Z-decay and ZGA genes. The *P* values were calculated by the two-sided Wilcoxon rank-sum test. The boxplots were generated with five percentile-based whiskers, arranged from top to bottom as follows: the 95th percentile, the 75th percentile, the median, the 25th percentile, and the 5th percentile; and the mean value indicated by a dot. *N* number of genes used in the data analysis. (D) Expression (left) and m⁶A status (right) dynamics of transcription cofactor coding genes. (E) Proportion of m⁶A+ transcription factors (left) and transcription cofactors (right) coding genes at each stage with different expression thresholds. (F) Expression and m⁶A marking status of genes essential for the lineage specification events in human early embryos. Top: gene expression, down: m⁶A status.

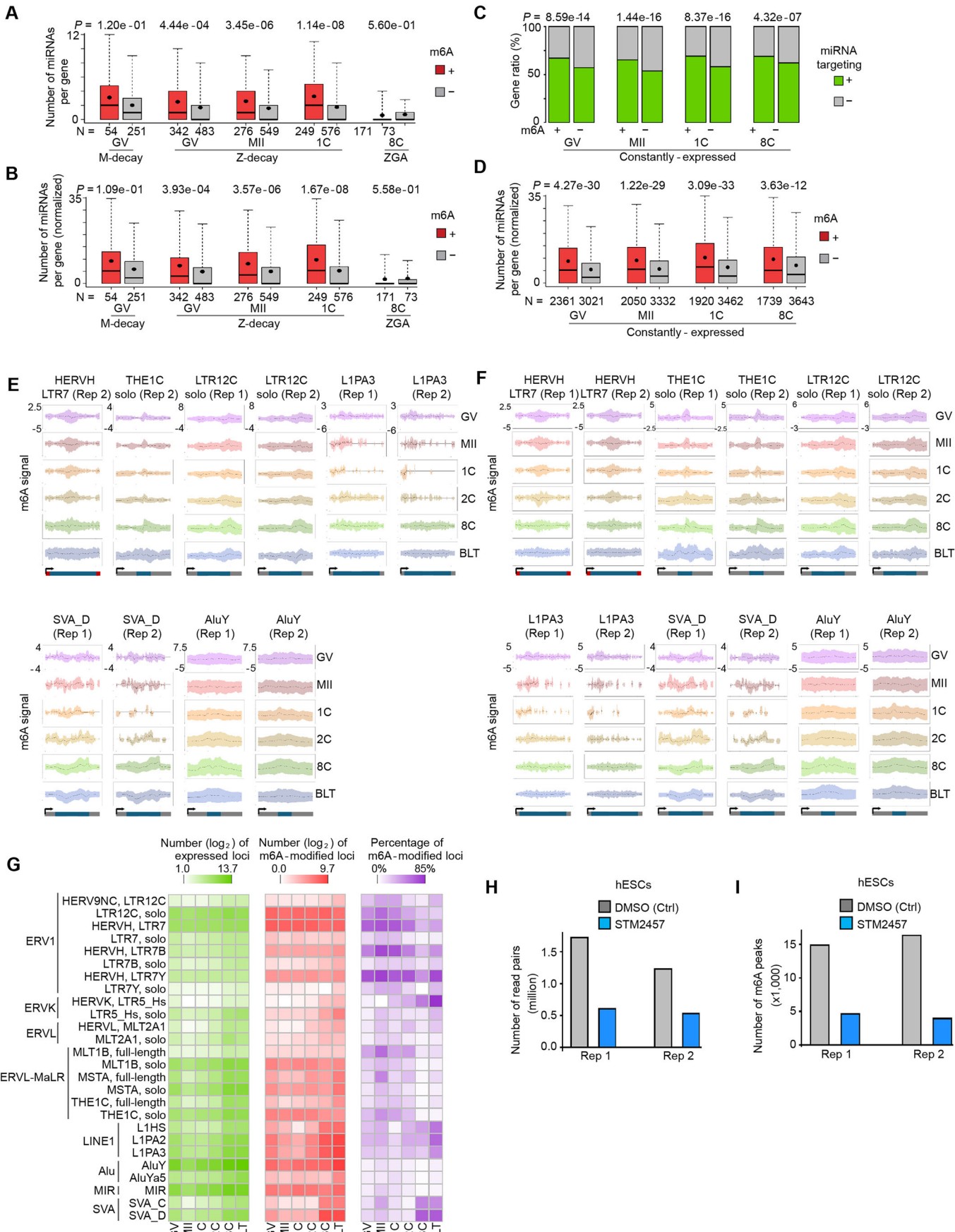

◄ **Figure EV6.  Comparison of miRNA targeting and translation efficiency between m⁶A modified and unmodified genes for M-decay, Z-decay and ZGA genes, as well as for constantly expressed genes; along with the analysis of m⁶A enrichment on retrotransposon-derived RNAs.**

(A) Comparison of the count of miRNAs targeting m⁶A modified versus unmodified mRNA genes for M-decay, Z-decay and ZGA genes. The *P* values were calculated by the two-sided Wilcoxon rank-sum test. *N* number of genes used in the data analysis. (B) Comparison of the normalized count of miRNAs targeting m⁶A modified versus unmodified mRNA genes for M-decay, Z-decay and ZGA genes. See "Methods" for the calculation of normalized mRNA counts. The *P* values were calculated by the two-sided Wilcoxon rank-sum test. *N* number of genes used in the data analysis. (C) Comparison of the proportion of mRNA genes targeted by miRNAs between m⁶A modified and unmodified genes for constantly expressed genes. The *P* values were calculated by the one-sided Fisher's exact test. (D) Comparison of the normalized count of miRNAs targeting m⁶A modified versus unmodified mRNA genes for constantly expressed genes. See "Methods" for the calculation of normalized mRNA counts. The *P* values were calculated by the two-sided Wilcoxon rank-sum test. *N* number of genes used in the data analysis. (E) Distribution of m⁶A signals along the full-length sequences of the representative retrotransposon subfamilies, based on unique assignment strategy. The mean (represented by the central black line) and standard deviation (represented by the band around the black line) across all loci were plotted. These plots are based on uniquely aligned reads. (F) Distribution of m⁶A signals along the full-length sequences of the representative retrotransposon subfamilies, based on a random assignment strategy. The mean (represented by the central black line) and standard deviation (represented by the band around the black line) across all loci were plotted. These plots are based on a random assignment strategy for those multiply aligned reads. (G) Heatmaps showing the number of expressed loci (left), the number of those with m⁶A among the expressed loci (middle), and the percentage of loci with m⁶A (right) for the representative retrotransposon subfamilies. These results are based on a random assignment strategy for those multiply aligned reads. (H) Comparison of the number of processed sequencing reads between control (DMSO) and treatment (STM2457) groups in human ESCs. (I) Comparison of the number of m⁶A peaks between control (DMSO) and treatment (STM2457) groups in human ESCs. The boxplots in (A, B, D) were generated with five percentile-based whiskers, arranged from top to bottom as follows: the 95th percentile, the 75th percentile, the median, the 25th percentile, and the 5th percentile; and the mean value indicated by a dot.

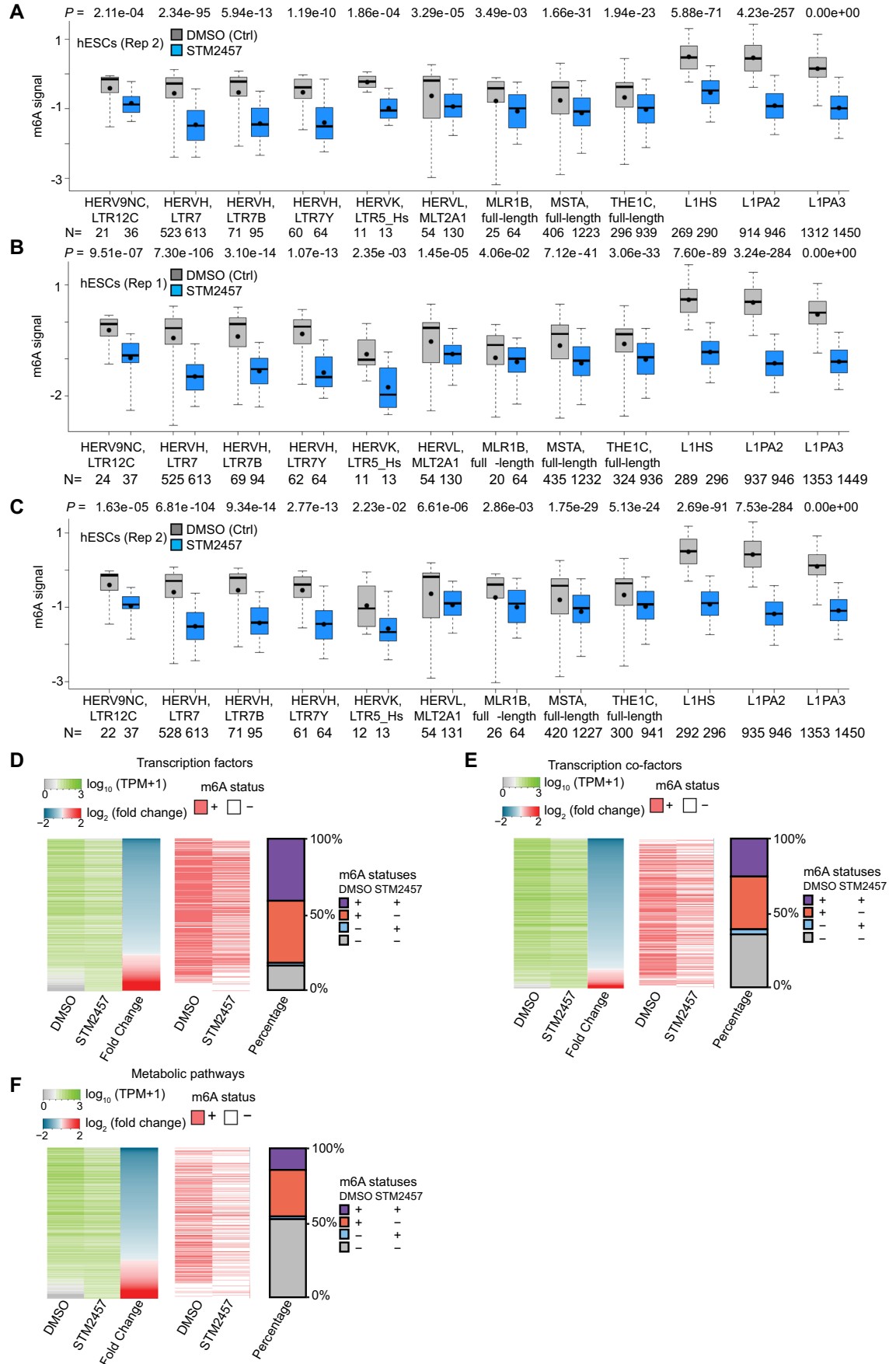

◀ **Figure EV7. Analysis of m⁶A profile changes after STM2457 treatment in hESC.**

(A) Comparison of the m⁶A signal on representative retrotransposon subfamilies between DMSO and STM2457 groups in hESCs (based on uniquely aligned reads from biological replicate 2). The *P* values were calculated by the two-sided Wilcoxon rank-sum test. *N* number of repeat loci used in the data analysis. (B, C) Comparison of the m⁶A signal on representative retrotransposon subfamilies between DMSO and STM2457 groups in hESCs (based on a random assignment strategy for those multiply aligned reads; see "Methods"). The (B) is biological replicate 1; and (C) is biological replicate 2. The *P* values were calculated by the two-sided Wilcoxon rank-sum test. *N* number of repeat loci used in the data analysis. (D–F) The expression level and fold change (left), m⁶A status (middle) and percentage (right) of transcription factor (D), transcription cofactors (E) and metabolic pathways (F) mRNAs. The boxplots in (A–C) were generated with five percentile-based whiskers, arranged from top to bottom as follows: the 95th percentile, the 75th percentile, the median, the 25th percentile, and the 5th percentile; and the mean value indicated by a dot.

