## [Peer Review File · The EMBO Journal]

The RNA m6A landscape during human oocyte-to-embryo transition

Yanjiao Li, Yunhao Wang, Aylin Gengiz, Kang-Xuan Jin, Blanca Corral Castroviejo, Xiaolin Lin, Marie Indahl, Rujuan Zuo, Trine Skuland, Madeleine Fosslie, Maria Biba, Xuechen Wu, Peter Fedorcsak, Magnar Bjørås, Adam Filipczyk, John Dahl, Gareth D. Greggains, Kin Au, and Arne Klungland

Corresponding author(s): Arne Klungland (arne.klungland@medisin.uio.no) , John Dahl (j.a.dahl@medisin.uio.no), Kin Au (kinfai@umich.edu), Gareth D. Greggains (g.d.greggains@medisin.uio.no)

Review Timeline:

Submission Date:	15th Oct 24
Editorial Decision:	26th Nov 24
Revision Received:	27th Feb 25
Editorial Decision:	28th Mar 25
Revision Received:	11th Apr 25
Accepted:	29th Apr 25

Editor: Cornelius Schneider

Transaction Report:

Dear Dr. Klungland,

Thank you for submitting your manuscript for consideration by the EMBO Journal.
The manuscript has now been seen by three referees and you can find their comments enclosed below.

As you will see from the reports, the reviewers appreciate the work, but also indicating a number of concerns. I think that the requested revisions are fair and reasonable and would therefore like to invite you to submit a revised manuscript addressing the concerns raised by all three referees. I am happy to discuss the revision in more detail via email or phone/videoconferencing should there be any additional questions.

We generally allow three months as standard revision time. As a matter of policy, competing manuscripts published during this period will not negatively impact on our assessment of the conceptual advance presented by your study. However, please contact me as soon as possible upon publication of any related work to discuss the appropriate course of action. Should you foresee a problem in meeting this three-month deadline, please contact us to arrange an extension.

When preparing your letter of response to the referees' comments, please bear in mind that this will form part of the Review Process File and will therefore be available online to the community. For more details on our Transparent Editorial Process, please visit our website: <https://www.embopress.org/page/journal/14602075/authorguide#transparentprocess>. Please also see the attached instructions for further guidelines on preparation of the revised manuscript.

Please feel free to contact me if you have any further questions regarding the revision. Thank you for the opportunity to consider your work for publication, and I look forward to your revision.

With best regards,

Cornelius Schneider

Cornelius Schneider, PhD
Editor
The EMBO Journal
c.schneider@embojournal.org

We realize that it is difficult to revise to a specific deadline. In the interest of protecting the conceptual advance provided by the work, we recommend a revision within 3 months (24th Feb 2025). Please discuss the revision progress ahead of this time with the editor if you require more time to complete the revisions. Use the link below to submit your revision:

Referee #1:

The manuscript presents the first comprehensive RNA m6A landscape during the human oocyte-to-embryo transition, utilizing the picoMeRIP-seq method, which enables profiling at the single-cell level, including individual human oocytes and early embryos. The key findings reveal fascinating dynamic changes in m6A modification levels across developmental stages, with significant differences observed between human and mouse embryos. The data quality is outstanding, making this an invaluable resource for the field of embryogenesis. To make this impressive manuscript even stronger, there are a few areas that could benefit from clarification or improvement.

1. The authors showed the number of m6A+ genes detected at each stage (Fig. 1b). The number of expressed genes varied greatly from oocyte, 1C, 8C (ZGA) and blastocyst. Detection of 5,000 m6A+ genes at oocyte stage, or at 8C during ZGA, could mean 90% or 50% of the total expressed genes being m6A+, respectively. Percentage of m6A+ genes at each stage would better represent the m6A dynamics along embryo development.
2. Blastocyst development involves trophoctoderm, primitive endoderm and epiblast specification from the totipotent 8C stage. Can the authors study the relationship between m6A modification and the lineage-specific gene sets/ markers in these critical lineages? This would broaden the role of m6A during human early embryo development.
3. Was the METTL3 inhibition experiment conducted in naïve or primed hESC? Primed hESC resembles epiblast post-implantation while this study covers pre-implantation embryos. For example, LINE-1 transcript levels vary significantly during blastocyst development/ implantation, and primed ESC has higher LINE-1 expression. Authors should consider to profile naïve hESC m6A to support m6A modification in modulating retrotransposon transcripts.
4. Can the authors show the changes of m6A profile changes in genes regulating transcription/ metabolic pathways in STM2457-treated hESC? Or other important pathways that could be regulated by m6A based on the embryo dataset analyses.
5. 7. Fig 1f, Fig S1f, Fig S2g : Need to indicate p-value.

Referee #2:

The authors presented the first single-cell RNA m6A profile of human oocytes and early embryos, as well as consistent and differential transcription and methylation characteristics corresponding to developmental stages in mice. Along with key events during early embryonic development, such as maternal RNA decay and zygotic genome activation, the regulation of m6A for different gene sets exhibits dynamic changes. Maternal decay RNAs tend to be modified by m6A and targeted by miRNA, while m6A modified RNAs with sustained expression exhibit higher translation efficiency. In addition, stage specific retrotransposons prefer to enrich m6A modification, especially in young subfamilies, and m6A inhibitors can erase a large amount of m6A from retrotransposons.

In summary, this study provides rich data resources for the characteristics and regulatory effects of m6A during early human embryonic development, which helps to deepen the its regulatory role in human early embryonic development. Besides, the manuscript is well-written and informative. These findings are intriguing in general. I still have some questions and suggestions that the authors could consider to address before the publication of the manuscript in EMBO Journal.

Major comments:

1. For the definition of m6A genes, the authors only have two IP samples for each developmental stage, but defined m6A+ if

m6A+ in any of two IP samples. Considering the authors depicted m6A signatures in single embryo using picoMeRIP-seq, if union strategy is used, whether it introduces too many heterogeneity modification features? Should the author consider using conservative features as the set for subsequent analysis.

2. Considering that the author used Pico technology to depict the modification features in a single embryo, if a union strategy is used, does it introduce too many single-cell heterogeneity modification features?

3. The author claimed that MD RNA tends to be modified by m6A with longer gene length, and is targeted by miRNA for degradation. But Fig. 3d shows that m6A modified ZGA genes also have longer gene length. How to explain the degradation and transcriptional activation of modified RNAs?

4. Fig. 4c showed higher translation efficiency for m6A-tagged MD RNAs, while lower translation efficiency for modified ZGA RNAs than unmodified ones. But Translation efficiency is closely related to RNA abundance. How to eliminate the effects of degradation and transcriptional activation on MD and ZGA genes?

5. For stage specific m6A-tagged retrotransposons, the authors only showed the profiles, while author should associate its regulatory targets and further elucidate its regulatory role in RNA metabolism processes such as transcription or translation during early human embryonic development.

Minor comments:

6. For RNA modification, the number of m6A should be superscripted as m6A.

7. During human early embryonic development, the author's explanation of the regulatory role of m6A on the transcription or translation of modified RNAs at different stages is not very clear. In addition to the transcription status of m6A related proteins at different stages, can authors combine protein data to elucidate the dynamics of m6A regulation in different stages or gene sets?

8. For the model diagram in Fig. 6, the regulatory effects of m6A modification on different gene sets are not clearly explained. It is recommended to reorganize it.

Referee #3:

Using their previously established picoMeRIP-seq method, the authors presented the first RNA m6A landscape of single human oocytes and early embryos, and analyzed the feature and changes of m6A profiles during early developmental stages of human embryos from different aspects. The results are informatics, yet there are still some questions need to be addressed.

1. Why does the 1C stage have fewer m6A-modified genes (3666) as compared with other stages, but the percentage of m6A-modified genes is higher?

2. Why is METTL3 not shown in Extended Fig 2d?

3. In Extended Fig 2f, how is the expected m6A peak numbers for each metagene region category calculated? The m6A peaks are enriched around stop codons in all reported experiments, how could the authors observe a more than 16 fold (\log_2 transformed ratio > 4) enrichment than expected in their samples? BTW, the explanation for top and bottom rows of Extended Fig 2f is reversed in the legend.

4. In lines 136 and 138, the authors should make it clear that the human/mouse specifically expressed genes shown in Fig 2b are only homologous genes. And what is the m6A status of non-homologous genes in each species? Similarly, for data in Fig. 2c, were they only derived from homologous genes?

5. It is kind of strange that the authors attributed the likelihood of having m6A modification to the numbers RRACH motifs in genes, as the formation of m6A has positional preference, which is not correlated with the number of RRACH motifs of a gene. Besides, gene groups with more RRACH motifs shown in Fig. 2d also have longer lengths in Fig. 3c, so the average RRACH motif density may not have significant difference.

6. Is the GO enrichment result shown in Fig. 2e consistent with the ones obtained from bulk cell m6A detection data? Why are heart development related genes expressed and have m6A modifications at these stages?

7. What percentage of genes (TPM > 50) be repeatedly detected in the two replicates of each sample? This could affect the accuracy of M-decay and Z-decay genes shown in Fig. 3, as the decay signal should be distinguished from the dropout noises.

8. The percentage reduction of m6A peaks in transcripts from retrotransposons (~47%) is lower than the overall loss of m6A peaks (73%), why is there such difference?

Minor points:

1. In Line 84, for the RRACH motif, the authors used the general nucleotide composition to explain the meaning of R and H, it would be more informative to reorder the nucleotides according to their appearance frequency in m6A motifs.

2. The authors used "highlight" twice in the sentence between lines 84-86, better to revise it.

3. There is some grammar error in sentences at lines 144-145.

4. In line 146, what does "human genes uniquely m6A+" mean?

Response to reviewers:

We are grateful to the editor and three reviewers for their evaluation of our work and their constructive feedback. The insightful comments have not only highlighted the potential value of this resource manuscript but have also identified critical areas for improvement.

Referee #1:

The manuscript presents the first comprehensive RNA m6A landscape during the human oocyte-to-embryo transition, utilizing the picoMeRIP-seq method, which enables profiling at the single-cell level, including individual human oocytes and early embryos. The key findings reveal fascinating dynamic changes in m6A modification levels across developmental stages, with significant differences observed between human and mouse embryos. The data quality is outstanding, making this an invaluable resource for the field of embryogenesis. To make this impressive manuscript even stronger, there are a few areas that could benefit from clarification or improvement.

Response: We sincerely thank reviewer for the thorough evaluation and positive recognition of our study as "an invaluable resource for the field of embryogenesis". We are pleased to address the insightful suggestions to strengthen the manuscript further.

1. The authors showed the number of m6A+ genes detected at each stage (Fig. 1b). The number of expressed genes varied greatly from oocyte, 1C, 8C (ZGA) and blastocyst. Detection of 5,000 m6A+ genes at oocyte stage, or at 8C during ZGA, could mean 90% or 50% of the total expressed genes being m6A+, respectively. Percentage of m6A+ genes at each stage would better represent the m6A dynamics along embryo development.

Response: We thank the reviewer's suggestion. We have now included the percentage of m⁶A+ genes at each stage in a new Extended Data Figure 2C. We also added the following sentence in the main text: "Among highly expressed genes (TPM ≥10), the percentage of m⁶A+ genes ranged from 32% to 44%, aligning with the overall trend observed in the number of m⁶A+ genes across the six stages".

2. Blastocyst development involves trophoctoderm, primitive endoderm and epiblast specification from the totipotent 8C stage. Can the authors study the relationship between m6A modification and the lineage-specific gene sets/ markers in these critical lineages? This would broaden the role of m6A during human early embryo development.

Response: We agree with the reviewer and have added additional analyses to include the m⁶A modification status of the lineage-specific marker genes for these three critical lineages in a new Extended Data Figure 8. We also added the following sentences in the main text: “The mature human blastocyst is comprised of the first three cell lineages of the embryo: trophoctoderm, epiblast and primitive endoderm. We further examined the m⁶A status on the lineage-specific marker genes across developmental stages and observed m⁶A occupancy on some master transcription regulators at the 8C and/or BLT stages, such as GATA2, GATA3 and TEAD3 (markers of trophoctoderm), SOX2 and NANOG (markers of epiblast), and PDGFRA and GATA6 (markers of primitive endoderm)”.

3. Was the METTL3 inhibition experiment conducted in naïve or primed hESC? Primed hESC resembles epiblast post-implantation while this study covers pre-implantation embryos. For example, LINE-1 transcript levels vary significantly during blastocyst development/implantation, and primed ESC has higher LINE-1 expression. Authors should consider to profile naïve hESC m⁶A to support m⁶A modification in modulating retrotransposon transcripts.

Response: We thank the reviewer for raising this critical point. During the revision period, we made rigorous attempts to achieve the suggested results; however, we encountered technical challenges.

Challenges with Naïve Conditions: Naïve conditions inhibit the primed-state FGF/ERK pathways while activating WNT, creating an unstable signaling equilibrium. Despite consistently testing the markers, we were unable to meet the stringent standard markers reported in the literature (PMID: 28429706, PMID: 26399828).

Experimental Efforts: We initiated experiments to inhibit METTL3 in naïve hESCs using reported culture conditions. However, we faced significant technical barriers: naïve hESCs demonstrated extreme sensitivity to small molecule inhibitors under naïve culture conditions. After several hours of treatment with METTL3 inhibitors, even with optimized dosage and timing, the phenotype of the cells changed dramatically compared to the naïve state, conferred greater hESCs viability, preventing us from continuing the three-day treatment procedure.

Collaboration Attempts: We reached out to other groups with expertise in naïve hESC culture but were unable to obtain compatible naïve state RNA samples with matched METTL3 inhibition within the revision timeframe.

We appreciate the reviewer's understanding of these experimental constraints. We are committed to continuing our work on m⁶A modification in human early embryos to further explore mechanistic insights in our follow-up studies.

4. Can the authors show the changes of m6A profile changes in genes regulating transcription/metabolic pathways in STM2457-treated hESC? Or other important pathways that could be regulated by m6A based on the embryo dataset analyses.

Response: We thank the reviewer for this suggestion. We have now included the analysis results in a new Extended Data Figure 13. We have also added a sentence in the main text: “Following STM2457 treatment, more than half of the genes involved in transcription and metabolic pathways lost m⁶A modification, yet their expression levels remained largely unchanged.”.

5. 7. Fig 1f, Fig S1f, Fig S2g: Need to indicate p-value.

Response: We thank the reviewer for this suggestion. We have now indicated all the p values in these figures.

Referee #2:

The authors presented the first single-cell RNA m6A profile of human oocytes and early embryos, as well as consistent and differential transcription and methylation characteristics corresponding to developmental stages in mice. Along with key events during early embryonic development, such as maternal RNA decay and zygotic genome activation, the regulation of m6A for different gene sets exhibits dynamic changes. Maternal decay RNAs tend to be modified by m6A and targeted by miRNA, while m6A modified RNAs with sustained expression exhibit higher translation efficiency. In addition, stage specific retrotransposons prefer to enrich m6A modification, especially in young subfamilies, and m6A inhibitors can erase a large amount of m6A from retrotransposons.

In summary, this study provides rich data resources for the characteristics and regulatory effects of m6A during early human embryonic development, which helps to deepen the its regulatory role in human early embryonic development. Besides, the manuscript is well-written and informative. These findings are intriguing in general. I still have some questions and suggestions that the authors could consider to address before the publication of the manuscript in EMBO Journal.

Response: We sincerely appreciate the reviewer for the valuable comments and we thank the reviewer's recognition of our work as "providing rich data resources" and the thoughtful suggestions to strengthen the biological implications.

Major comments:

1. For the definition of m6A genes, the authors only have two IP samples for each developmental stage, but defined m6A+ if m6A+ in any of two IP samples. Considering the authors depicted m6A signatures in single embryo using picoMeRIP-seq, if union strategy is used, whether it introduces too many heterogeneity modification features? Should the author consider using conservative features as the set for subsequent analysis.

Response: We appreciate the reviewer's insightful feedback and agree that using a union strategy is generally considered less conservative in data analysis. We chose the union strategy for the following reasons:

1) *Technical robustness:* The intersection strategy is often employed to address technical variations. A single hESC contains approximately 10 to 30 picograms (pg) of total RNA, while a single human oocyte or early embryo contains about 200 to 500 pg of total RNA. In our study, as demonstrated in Extended Data Figure 1, even when downscaled to 10 hESCs, the correlation/consistency between two replicates remains excellent, indicating the robustness of our methodology. Given that the total

RNA in a single human oocyte or embryo exceeds that of 10 hESCs, we anticipate achieving superior results without the need for an intersection strategy to address technical noise.

- 2) *Comparability across stages*: To maintain statistical rigor and ensure comparability between stages, we selected two biological replicates from each stage for downstream analyses, even though more than two biological replicates are available for some stages (Extended Data Figure 2A). This approach allows for rigorous stage comparisons. Applying an intersection strategy would result in fewer m⁶A+ genes for stages with more biological replicates.
- 3) *More targets for downstream functional studies*: Existing omics studies of mouse/human early embryos commonly used a union strategy (pooling replicates together) to explore biological meanings and dynamic omics profiles over development [PMID: 28621329; PMID: 29915357; PMID: 38238450; PMID: 39080410]. In our study, employing a union strategy also provides more m⁶A+ gene targets for downstream studies to investigate the roles of m⁶A in regulating mammalian early embryonic development.

To emphasize the heterogeneity present in our single-oocyte/embryo data, we added the following sentence in the Discussion section: “*Due the limited availability of human embryos, we were unable to conduct large-scale single-embryo analyses to explore heterogeneity or experimentally validate these intriguing questions.*”

2. Considering that the author used Pico technology to depict the modification features in a single embryo, if a union strategy is used, does it introduce too many single-cell heterogeneity modification features?

Response: We sincerely thank the reviewer for pointing out the potential heterogeneity of our single-oocyte/embryo data. As described in the first/above concern, we anticipate that this heterogeneity is more likely to reflect biologically meaningful patterns rather than technical randomness. Besides, exploring heterogeneity across different single human embryos requires more replicates to achieve statistical significance. This aspect will be a valuable focus of future research, and we are actively pursuing it in our follow-up studies.

3. The author claimed that MD RNA tends to be modified by m⁶A with longer gene length, and is targeted by miRNA for degradation. But Fig. 3d shows that m⁶A modified ZGA genes also have longer gene length. How to explain the degradation and transcriptional activation of modified RNAs?

Response: We sincerely appreciate the reviewer's insightful question regarding the dual roles of m⁶A in RNA degradation and transcriptional activation. The referee is correct, our analysis shows that longer MD and ZGA genes tend to exhibit higher m⁶A levels, this is indeed a critical point requiring further explanation, and we propose the

following interpretations based on recent publications and our data. Advances in understanding the mechanism of m⁶A deposition suggest that m⁶A functions akin to a hardware feature that is added by default [PMID: 36550132; PMID: 36599352; PMID: 36705538]. Exon junction complexes (EJCs) have been identified as m⁶A 'suppressor' that protect exon junction-proximal RNA within coding sequences from being methylated. This EJC protection is typically sufficient to suppress m⁶A deposition in average-length internal exons but not in long internal and terminal exons. Longer genes tend to have more exons and splicing events, increasing the availability of RRACH motifs in exonic regions. The activity of EJCs helps define where m⁶A marks are placed along the transcript, especially in longer genes where more exons allow for more regions susceptible to m⁶A deposition. This unified model provides basic insights into m⁶A for its selective deposition. It encourages us to explore whether the unified model varies depending on the context in human early embryos. Firstly, to compare the gene length and motif number of m⁶A-modified genes as a form of quality control, and check if our data aligns with the unified model. Secondly, while the groundbreaking unified model was developed in cultured cells, we were interested to see if a different pattern exists during the human oocyte to embryo transition. Our results indicate that m⁶A-modified genes do tend to be longer and contain more motifs than unmodified ones, even within this specific biological context. There is no inherent difference in RNA degradation and transcriptional activation of modified RNAs based on gene length alone. We have added a discussion on this point in Lines 325 – 333 to further clarify these observations. Previous studies have demonstrated that m⁶A promotes maternal RNA degradation and is indispensable post-zygotic genome activation in both zebrafish and mice. These functions are largely mediated by m⁶A writer and reader proteins during oogenesis and early embryogenesis [PMID: 32094512, PMID: 28867294, PMID: 31406667, PMID: 33658714, PMID: 35606490]. However, the precise regulatory mechanisms and molecular dynamics of m⁶A on RNA in oocytes and early embryos remain poorly characterized due to limited biological material availability in early developmental stages. The detailed regulatory model is largely ambiguous. The field and our research group are still actively investigating this mechanism.

4. Fig. 4c showed higher translation efficiency for m⁶A-tagged MD RNAs, while lower translation efficiency for modified ZGA RNAs than unmodified ones. But Translation efficiency is closely related to RNA abundance. How to eliminate the effects of degradation and transcriptional activation on MD and ZGA genes?

Response: We apologize for any confusion regarding the significance of the p-values. As stated in the main text, we found that m⁶A+ Z-decay genes had relatively higher translation efficiency than unmodified genes at the MII and 1C stages. However, there was no significant association between m⁶A modification and mRNA translation efficiency for M-decay and ZGA genes (Fig. 4C). To improve clarity, we have now

highlighted significant p-values in red color for better readability in Figure 4C. For the calculation of translation efficiency, we used data from a published study that combined Ribo-lite with Smart-seq2, a low-input RNA-seq method [PMID: 36074823]. This combined approach (as below figure showed), termed Ribo-RNA-lite (R2-lite), allows transcriptome and translome co-profiling by splitting lysed samples. Ribo-lite provides a snapshot of active translation by sequencing RNA fragments protected by ribosomes, and when combined with RNA-seq, it reveals which mRNAs are being actively translated and at what levels. Translation efficiency was normalized to mRNA abundance as described in Zou et al. 2022 [PMID: 36074823]. As illustrated in the figure below, translational efficiency (TE) was calculated as the ratio of Ribo-lite and mRNA-seq (FPKM+1/FPKM+1), ensuring that changes in RNA stability or transcription rates do not confound TE measurements. This method is unrelated to RNA stability or transcriptional activity, as TE represents translational activity per normalized mRNA molecule, thereby allowing us to focus solely on translation efficiency.

Schematic of R2-lite and translational efficiency formula

5. For stage specific m⁶A-tagged retrotransposons, the authors only showed the profiles, while author should associate its regulatory targets and further elucidate its regulatory role in RNA metabolism processes such as transcription or translation during early human embryonic development.

Response: We appreciate the reviewer's suggestion to investigate the regulatory targets and roles of stage-specific m⁶A-tagged retrotransposons. However, there are inherent limitations in our picoMeRIP-seq method that hinder us from achieving such detailed insights. Specifically, we sonicated rRNA-depleted RNAs into short fragments and employed m⁶A antibody immunoprecipitation to isolate m⁶A-modified RNA fragments. Due to the repetitive nature of retrotransposons, sequencing and mapping can result in the loss of information about these elements and their neighboring mRNAs, impeding our ability to explore how retrotransposons may regulate adjacent RNAs. While the latest long-read sequencing technologies present the potential to capture m⁶A-modified retrotransposons along with adjacent mRNAs by generating reads that span entire retrotransposon elements and their flanking regions, our current short-read approach is inadequate for investigating how m⁶A-modified retrotransposons might

influence transcription and translation. Additionally, techniques such as CLIP-seq could uncover interactions between retrotransposon-derived RNAs and m⁶A readers or translation-related factors, but these also require long-read sequencing. Applying long-read sequencing and CLIP-seq would demand thousands of human embryos, which is not feasible given the scarcity of available human embryonic material. We acknowledge these challenges and recognize that the field of human embryo research is actively seeking alternative methods to address these limitations in future studies.

Minor comments:

6. For RNA modification, the number of m⁶A should be superscripted as m⁶A.

Response: We thank the reviewer for the suggestion and have corrected them in the manuscript.

7. During human early embryonic development, the author's explanation of the regulatory role of m⁶A on the transcription or translation of modified RNAs at different stages is not very clear. In addition to the transcription status of m⁶A related proteins at different stages, can authors combine protein data to elucidate the dynamics of m⁶A regulation in different stages or gene sets?

Response: We thank the reviewer for the suggestion. We have obtained the protein data (mass spectrometry) from a recent study [PMID: 39855199] and analyzed the protein expression of m⁶A-related genes. The results have been added into a new Extended Data Figure 3B and the main text (Line 105-108).

8. For the model diagram in Fig. 6, the regulatory effects of m⁶A modification on different gene sets are not clearly explained. It is recommended to reorganize it.

Response: We thank the reviewer for the insightful suggestion. The model diagram in Figure 6 has been reorganized to emphasize the conserved m⁶A modification patterns between humans and mice, along with m⁶A stage-specific regulatory roles in early human embryogenesis presented in our study. Additionally, a bottom timeline has been integrated to illustrate the dynamic association of m⁶A with developmental stages.

Referee #3:

Using their previously established picoMeRIP-seq method, the authors presented the first RNA m6A landscape of single human oocytes and early embryos, and analyzed the feature and changes of m6A profiles during early developmental stages of human embryos from different aspects. The results are informatics, yet there are still some questions need to be addressed.

Response: We appreciate the reviewer's valuable suggestions. We have carefully incorporated the reviewers' valuable feedback to improve the quality and clarity of the manuscript.

1. Why does the 1C stage have fewer m6A-modified genes (3666) as compared with other stages, but the percentage of m6A-modified genes is higher?

Response: We thank the reviewer's question. In the initial manuscript, we did not present the percentage of m⁶A-modified genes at different stages. We have now included this information in a new Extended Data Figure 2C and in the main text (Line 99-108). Our results show that the percentage of m⁶A-modified genes at the 1C stage is also lower as compared with other stages. Regarding the 1C stage having fewer m⁶A-modified genes, we attribute this to the oocyte-to-zygote transition. During this period, significant degradation of maternal RNAs occurs, while zygotic mRNAs have not yet been synthesized in substantial amounts.

2. Why is METTL3 not shown in Extended Fig 2d?

Response: We apologize for the unclear labeling. We have now enlarged the font in Extended Fig 3A and placed the labels above the figure for easier readability.

3. In Extended Fig 2f, how is the expected m6A peak numbers for each metagene region category calculated? The m6A peaks are enriched around stop codons in all reported experiments, how could the authors observe a more than 16 fold (log₂ transformed ratio > 4) enrichment than expected in their samples? BTW, the explanation for top and bottom rows of Extended Fig 2f is reversed in the legend.

Response: We thank for the reviewer's question. The expected number of peaks is calculated based on the length proportion of different genomic regions (i.e., 5'UTR, CDS, 3'UTR, stop codon, exon, intron, intergenic) relative to the total length of reference genome hg38.

For example:

(1) If the proportion of stop codon relative to the total length of reference genome is 0.5%, and

(2) The total number of m⁶A peaks identified in the experimental data is 1000, with 200 peaks observed in (assigned to) the stop codon region,

The expected number of peaks for the stop codon region is calculated as:

$$\text{Expected peaks} = \text{Total peaks} \times \text{Length proportion} = 1000 \times 0.5\% = 5$$

The enrichment score is then calculated as:

$$\text{Enrichment score} = \log_2(\text{observed peaks} / \text{expected peaks}) = \log_2(200/5) = 5.32.$$

To clarify, we have added this information into the Methods section (Line 961 - 965). We have also corrected the legends of Extended Data Figure 2F.

4. In lines 136 and 138, the authors should make it clear that the human/mouse specifically expressed genes shown in Fig 2b are only homologous genes. And what is the m⁶A status of non-homologous genes in each species? Similarly, for data in Fig. 2c, were they only derived from homologous genes?

Response: We apologize for the unclear writing and labeling. We have now revised the main text and figure legends/labels for improved readability. Regarding the m⁶A status of non-homologous genes in each species, we have included new results in a new Extended Data Figure 4B and added the following sentence in the main text: "For the non-homologous genes between the two species, an average of 39% were m⁶A+ across all stages in human, compared to 49% in mouse, with a notably low fraction of m⁶A modification at the BLT stage for both human (20%) and mouse (35%).". Additionally, we confirm that the data in Figure 2C were derived from homologous genes.

5. It is kind of strange that the authors attributed the likelihood of having m⁶A modification to the numbers RRACH motifs in genes, as the formation of m⁶A has positional preference, which is not correlated with the number of RRACH motifs of a gene. Besides, gene groups with more RRACH motifs shown in Fig. 2d also have longer lengths in Fig. 3c, so the average RRACH motif density may not have significant difference.

Response: We sincerely appreciate the reviewer's insightful question. Upon analyzing the average RRACH motif density, we found no significant difference, as the reviewer predicted. These new findings have been added to a new Extended Data Figure 5. Our results align with recent models of m⁶A deposition, which suggest that m⁶A is typically added at RRACH motifs but excluded from regions near splice sites [PMID: 36550132; PMID: 36599352; PMID: 36705538]. Advances in understanding m⁶A deposition mechanisms indicate that m⁶A acts like a default hardware feature. Exon junction complexes (EJCs) have been identified as m⁶A suppressors, protecting RNA near exon

junctions within coding sequences from methylation. This protection is generally adequate to prevent m⁶A deposition in average-length internal exons, but not in long internal and terminal exons. Longer genes typically have more exons, which increases the availability of RRACH motifs in exonic regions. Moreover, spliced constructs tend to strongly suppress methylation of sequences unless they are placed within the last exon, due to the absence of splice complexes at the ends of final exons.

6. Is the GO enrichment result shown in Fig. 2e consistent with the ones obtained from bulk cell m⁶A detection data? Why are heart development related genes expressed and have m⁶A modifications at these stages?

Response: Unfortunately, there is no bulk data available. Our study provides the first m⁶A profiles in human early embryos. GO analysis identifies biological processes that are statistically overrepresented in a gene set. However, since genes can be involved in multiple biological processes, a few GO terms might be significantly enriched even if they seem biologically irrelevant to the specific stage under study. This phenomenon has also been observed in GO analyses of previous early embryo studies. For example, the GO term "regulation of epithelial-to-mesenchymal transition" was identified as statistically significant in m⁶A-modified maternal decay genes in mouse early embryos [PMID: 35606490], and the GO term "muscle system process" was presented in human early embryos [PMID: 23934149].

7. What percentage of genes (TPM > 50) be repeatedly detected in the two replicates of each sample? This could affect the accuracy of M-decay and Z-decay genes shown in Fig. 3, as the decay signal should be distinguished from the dropout noises.

Response: In generating the heatmap, we followed the standard procedure for studies on human early embryos by merging the two biological replicates. For the bioinformatics pipeline, we cannot treat them as separate samples since we require 'input' RNA-seq data for m⁶A peak identification. To achieve optimal transcriptome coverage in single oocyte/embryo 'input', we combined 10% from each biological replicate to create a pooled 'input' sample, resulting in only one replicate for 'input'. This is why we used a TPM threshold of 50, as it significantly reduces noise. Furthermore, we set a threshold of TPM ≥ 10 for defining M-decay and Z-decay genes. As shown in attached revision Table 1, the number of genes with TPM ≥ 50 is comparable across the six stages, ensuring consistency and reliability in our analysis.

8. The percentage reduction of m⁶A peaks in transcripts from retrotransposons (~47%) is lower than the overall loss of m⁶A peaks (73%), why is there such difference?

Response: We appreciate the reviewer for raising this interesting question. Retrotransposon transcripts and mRNAs differ significantly in structure and function. Retrotransposons are repetitive elements, which might make them less sensitive to m⁶A modification machinery inhibitors used in our study. This insensitivity could be due to structural or sequence-specific characteristics that offer protection against m⁶A peak removal. Additionally, retrotransposons may have crucial roles that necessitate maintaining a certain level of m⁶A modifications, even under inhibitor conditions. The transcriptional regulation of retrotransposons may also differ inherently, resulting in a slower or less pronounced reduction in m⁶A peaks compared to other mRNAs when exposed to inhibitors. We have included this discussion in Lines 307 – 310.

Minor points:

1. In Line 84, for the RRACH motif, the authors used the general nucleotide composition to explain the meaning of R and H, it would be more informative to reorder the nucleotides according to their appearance frequency in m⁶A motifs.

Response: We thank the reviewer for the suggestion and have reordered the motif.

2. The authors used "highlight" twice in the sentence between lines 84-86, better to revise it.

Response: We thank the reviewer for the suggestion and have revised the sentence.

3. There is some grammar error in sentences at lines 144-145.

Response: We have corrected the error.

4. In line 146, what does "human genes uniquely m⁶A+" mean?

Response: We thank the reviewer for noticing this and have corrected it in the manuscript at lines 162-163.

Dear Dr. Klungland,

thank you for submitting a revised version of your manuscript. Your study has now been seen by all original referees, who find that their previous concerns have been addressed and now recommend publication of the manuscript. There remain only a few mainly editorial points that have to be addressed before I can extend formal acceptance of the manuscript:

- Please double-check to make sure to all relevant funding information in the manuscript is also entered into our submission system ("More Funders" option in eJP)
- On the abstract page of the manuscript, please include 4-5 general keyword terms to enhance searchability.
- Please adjust the format of the reference list and of the in-text citations according to EMBO Journal format (alphabetical order, author name et al + year.../up to 10 author names in the reference list before et al / please refer to our Guide to Authors for additional information on EMBO J reference format).
- Please rename the Conflict of Interest section into "Disclosure and Competing Interests Statement", in accordance with our updated Guide to Authors (<https://www.embopress.org/competing-interests>)
- As we are switching from a free-text author contribution statement towards a more formal statement based on Contributor Role Taxonomy (CRediT) terms, please remove the present Author Contribution section and instead specify each author's contribution(s) directly in the Author Information page of our submission system during upload of the final manuscript. See <https://casrai.org/credit/> for more information.
- Please adjust the in-text callouts for individual figures and figure panels: All callouts should be listed sequentially and there are callouts, but no such panels: Fig. 2F-G; 6A-C; missing callouts for supplementary figure panels.
- Please provide either a "Yes" or a "Not Applicable" answer to each one of the questions in your Author Checklist (<https://www.embopress.org/pb-assets/embo-site/EMBO%20Press%20Author%20Checklist-1642513524327.xlsx>). In the last column of this checklist, only the sections of the manuscript where the relevant information can be found should be listed (the information per se should be included in the main manuscript file).
- Please upload figures as individual Figure files with legends placed in ms file below the References; Regarding the supplementary figures, you can upload up to 5 (7) as EV Figures, in which case they need to be uploaded as separate Figure files as well, with their legends in the manuscript file, after the main figure legends. The nomenclature for EV figures should be Figure EVx in EV figure legends and callouts.
- DATASET EV LEGENDS: nomenclature should be Dataset EV1-EVx with the corresponding callouts; legends should be removed from ms file
- APPENDIX 1 FILE WITH ToC: The remaining figures should be compiled in one PDF file labeled "Appendix" with their legends. Appendix file needs to be in PDF format; title page should contain "Appendix for + ms title" and ToC with the page numbers for the listed items; nomenclature should be Appendix Figure Sx and Appendix Table Sx throughout ms and Appendix PDF
- Please provide the Reagent and Tools Table. For more information, please check <https://www.embopress.org/page/journal/14602075/authorguide#structuredmethods> and download the template for Reagent Table
- Please provide suggestions for a short 'blurb' text prefacing and summing up the conceptual aspect of the study in two sentences (max. 250 characters), followed by 3-5 one-sentence 'bullet points' with brief factual statements of key results of the paper; they will form the basis of an editor-written 'Synopsis' accompanying the online version of the article. Please also provide an altered synopsis image, making sure that the aspect ratio conforms to our website's format - it should be exactly 550 pixels wide and between 300-600 pixels high.
- Figure Legends (main + EV):
 1. Please indicate the statistical test used for data analysis in the legends of figures 2E, 3F, 4B-D.
 2. Please note that the box plots need to be defined in terms of minima, maxima, centre, bounds of box and whiskers, and percentile in the legends of figures 2C, D; 3D, E; 4B, C, D, E; 4D
 3. Please note that information related to n is missing in the legends of figures 2C, D; 3D, E; 4B, C, D, E; 4D
- Table 1 should be renamed to Table EV1 with the appropriate callout; legend should be removed from ms file
- Section order should be corrected: Title page - Abstract & Keywords - Introduction - Results - Discussion - Methods - Data Availability - Acknowledgements - Disclosure and Competing Interests Statement - References - Figure Legends - Table(s) - Expanded View Figure Legends.

With best regards,

Cornelius Schneider

Editor | The EMBO Journal
c.schneider@embojournal.org

We realize that it is difficult to revise to a specific deadline. In the interest of protecting the conceptual advance provided by the work, we recommend a revision within 3 months (26th Jun 2025). Please discuss the revision progress ahead of this time with the editor if you require more time to complete the revisions. Use the link below to submit your revision:

Referee #1:

The authors have done a great job in addressing my concerns from the first review. I support its publication in EMBO journal.

Referee #2:

The authors have addressed most of my concerns. I recommend its publication.

Referee #3:

The authors have addressed all my questions, I support the manuscript for publication.

All editorial and formatting issues were resolved by the authors.

Dear Prof. Klungland,

I am pleased to inform you that your manuscript has been accepted for publication in the EMBO Journal.

Yours sincerely,

Cornelius Schneider, PhD
Editor
The EMBO Journal
c.schneider@embojournal.org
